# ATP-induced asymmetric pre-protein folding as a driver of protein translocation through the Sec machinery

Robin A Corey[1†‡], Zainab Ahdash[2†], Anokhi Shah[3], Euan Pyle[2,4], William J Allen[1], Tomas Fessl[5], Janet E Lovett[3]*, Argyris Politis[2]*, Ian Collinson[1]*

[1]School of Biochemistry, University of Bristol, Bristol, United Kingdom; [2]Department of Chemistry, King's College London, London, United Kingdom; [3]SUPA School of Physics and Astronomy and BSRC, University of St Andrews, Scotland, United Kingdom; [4]Department of Chemistry, Imperial College London, London, United Kingdom; [5]University of South Bohemia in Ceske Budejovice, České Budějovice, Czech Republic

**\*For correspondence:**
jel20@st-andrews.ac.uk (JEL);
argyris.politis@kcl.ac.uk (AP);
ian.collinson@bristol.ac.uk (IC)

[†]These authors contributed equally to this work

**Present address:** [‡]Department of Biochemistry, University of Oxford, Oxford, United Kingdom

**Competing interests:** The authors declare that no competing interests exist.

**Abstract** Transport of proteins across membranes is a fundamental process, achieved in every cell by the 'Sec' translocon. In prokaryotes, SecYEG associates with the motor ATPase SecA to carry out translocation for pre-protein secretion. Previously, we proposed a Brownian ratchet model for transport, whereby the free energy of ATP-turnover favours the directional diffusion of the polypeptide (Allen et al., 2016). Here, we show that ATP enhances this process by modulating secondary structure formation within the translocating protein. A combination of molecular simulation with hydrogendeuterium-exchange mass spectrometry and electron paramagnetic resonance spectroscopy reveal an asymmetry across the membrane: ATP-induced conformational changes in the cytosolic cavity promote unfolded pre-protein structure, while the exterior cavity favours its formation. This ability to exploit structure within a pre-protein is an unexplored area of protein transport, which may apply to other protein transporters, such as those of the endoplasmic reticulum and mitochondria.

## Introduction

The encapsulation and compartmentalisation of cells has necessitated the evolution of machineries that conduct proteins across membranes, including for protein secretion and organellar import. Usually, protein transport occurs before the nascent protein has folded. This paper explores how the protein folding process per se may be exploited to drive protein translocation.

The bulk of protein secretion and membrane protein insertion is conducted by the ubiquitous Sec translocon. In bacteria, this comprises SecY, SecE and usually SecG, with the protein-conducting pore running through the centre of SecY. This complex can associate with either the ribosome for co-translational protein translocation (*Blobel and Dobberstein, 1975*) – the main pathway for nascent membrane protein insertion in bacteria (*Müller et al., 2001*; *Jungnickel et al., 1994*; *Ulbrandt et al., 1997*) – or with the motor protein SecA for post-translational secretion of pre-proteins (*Hartl et al., 1990*), which contain an N-terminal cleavable signal sequence. In the latter case, the fully synthesised pre-protein is maintained in an unfolded conformation by chaperones, such as SecB, and SecA itself (*Hartl et al., 1990*; *Arkowitz et al., 1993*). The protein must then fold during or after the translocation process.

Post-translational translocation of the unfolded pre-protein occurs through a contiguous channel formed through SecA and SecY (*Figure 1A–C*; *Zimmer et al., 2008*; *Van den Berg et al., 2004*; *Li et al., 2016*; *Bauer and Rapoport, 2009*). The first step in this process is the ATP-dependent

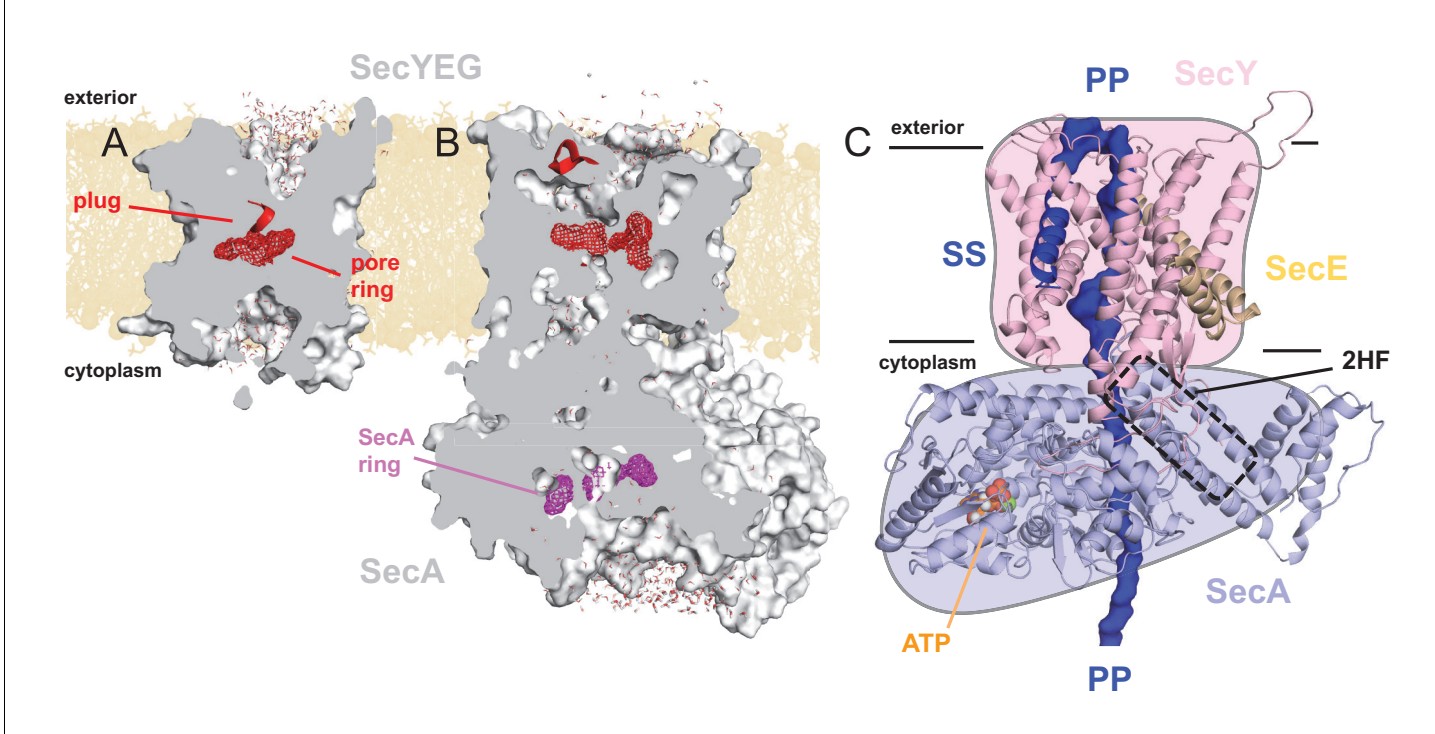

**Figure 1.** Structures of SecYEG and SecA. Interior views of (**A**) SecYEβ (PDB 1RHZ; *Van den Berg et al., 2004*) and (**B**) SecYEG-SecA (PDB 3DIN; *Zimmer et al., 2008*) showing the cavities through the channel, with the protein in grey surface, the pre-protein pore constrictions in red (SecY) or purple (SecA) mesh, and the SecY plug in red helix. The image was produced by embedding the crystal structures in a POPC membrane, solvating with explicit waters and allowing the non-heavy atoms to relax through restrained molecular dynamics (MD) over 4 ps. Degree of solvation and water density in the channel to be considered for illustrative purposes only. (**C**) Cartoon representation of SecA-SecYEG with an engaged pre-protein, modelled from PDB 5EUL (*Li et al., 2016*). SecY is shown in light pink, SecE orange, and SecA light-blue, with the 2HF highlighted. The unfolded pre-protein is shown in dark blue surface, labelled 'PP', and the signal sequence as blue cartoon ('SS'). The ATP analogue is coloured as orange (phosphate), blue (nitrogen) and red (oxygen) spheres. The approximate position of the membrane is marked. The cartoon is overlaid on a coloured schematic, used throughout the manuscript.

The online version of this article includes the following figure supplement(s) for figure 1:

**Figure supplement 1.** Details on remodelling of the SecA-SecYE-PP complex.

'initiation' phase, whereby the pre-protein is targeted *via* its cleavable N-terminal signal sequence to SecA, and subsequently to the SecYEG complex (*Fessl et al., 2018*). Transfer of the signal sequence to SecY unlocks the protein-channel, enabling the intercalation of the rest of the pre-protein (*Corey et al., 2016*; *Hizlan et al., 2012*). Next, the pre-protein is fed through the SecY channel in a process driven by both ATP and the proton-motive-force (PMF) (*Brundage et al., 1990*), with each mechanism requiring specific interactions with cardiolipin (*Gold et al., 2010*; *Corey et al., 2018*; *Hendrick and Wickner, 1991*). Finally, the signal sequence is cleaved and the pre-protein is either folded or trafficked onwards.

The SecY pore is constricted centrally by a ring of six hydrophobic residues (*Van den Berg et al., 2004*) (*Figure 1A*; red mesh), which is flanked on the cytoplasmic and exterior sides by large solvated cavities, resulting in an hourglass-like configuration (*Van den Berg et al., 2004*) (*Figure 1A*). This pore ring is capped by the so-called 'plug' motif (*Van den Berg et al., 2004*), which sits on the external face of the pore (*Figure 1A*; red helix), and helps to seal the channel against undesirable ion flux (*Park and Rapoport, 2011*). Upon SecA binding, SecY rearranges to form a more open state, involving the relocation of the plug (*Zimmer et al., 2008*), likely controlled by the ATPase activity of SecA (*Fessl et al., 2018*; *Allen et al., 2016*). This state still contains solvated cytoplasmic and exterior cavities, but with a profoundly changed shape of the channel (*Figure 1B*).

Recently, additional information on the translocation process has come from a pseudo-translocation state captured in a crystal structure of *Geobacillus thermodentrificans* SecYE bound to *Bacillus*

*subtilis* SecA (*Li et al., 2016*); remodelled for *Figure 1C*. This structure was determined by engineering a region of pre-protein into the SecA two helix finger (2HF), such that it resides within the SecY channel. Amongst many observations, the data reveal that the central pore of SecY tightly clasps the pre-protein (*Figure 1—figure supplement 1A*).

Previously, we proposed a Brownian ratchet mechanism for ATP-driven protein secretion, where stochastic pre-protein diffusion is biased into directional force by the action of the SecA ATPase (*Allen et al., 2016*). We demonstrated that ATPase activity was able to regulate the open (ATP) and closed (ADP) states of the SecY channel; thus allowing/preventing passage of specific regions of pre-protein, such as large or charged residues, or small helical regions of secondary structure. The SecA 2HF can sense these regions, and communicate their presence back to SecA to trigger nucleotide exchange, thereby providing the means to bias Brownian motion (see Figure 8 and Video 1 in *Allen et al., 2016*).

Here, we extend this model to include structural changes within the channel and translocating pre-protein. We employ atomistic molecular dynamics (MD) simulation on a remodelled version of the pseudo-translocation crystal structure, along with electron paramagnetic resonance (EPR) spectroscopy and hydrogen deuterium exchange mass spectrometry (HDX-MS) on *E. coli* SecA-SecYEG. Together, the data reveal structural changes between the cytoplasmic and exterior cavities of SecY during the SecA ATPase cycle. These changes are instigated by a widening of the cytoplasmic cavity in the ATP-bound state. In the simulation data, this widening results in a reduced degree of pre-protein secondary structure when compared to the exterior cavity. We show this asymmetry is strongly diminished in the ADP-bound state, suggesting pre-protein transport is, in part, driven by ATP-dependent control of secondary structure formation.

## Results

### Asymmetric secondary structure formation of translocon-engaged pre-protein

To investigate the structural changes in the pre-protein during the translocation process, we remodelled the pseudo-translocation state crystal structure (PDB 5EUL [*Li et al., 2016*]) into a physiological complex (i.e. a non-fusion protein) containing SecA, SecYE and 76 residues of unfolded pre-protein, hereafter referred to as 'SecA-SecYE-PP'; see Materials and methods section for full modelling detail (*Figure 1—figure supplement 1B–C* and *Table 1*). This structure was built into a solvated lipid bilayer, and simulated over 1 μs with SecA occupied by either ADP or ATP. The simulations were stable within the core SecY region (*Figure 2—figure supplement 1*) and, crucially, the SecY pore remained tightly formed around the bound pre-protein (*Figure 2—figure supplement 2A*).

A range of hydrogen-bonded secondary structure motifs are observed within the SecY channel, mainly α-helix in the ATP simulation and $3_{10}$-helix in the ADP simulation (*Figure 2—figure supplement 2B*). Strikingly, however, most of this secondary structure appears to form in the exterior cavity, with little on the cytosolic side of the pore ring (e.g. *Figure 2A*).

Deformation analysis, which reports on local conformational flexibility in structural ensembles (see Materials and methods including *Figure 2—figure supplement 3A* for details), reveals a high degree of pre-protein perturbation within SecY, particularly in the cytoplasmic cavity (*Figure 2B*). As the pre-protein is observed to fold primarily in the exterior cavity, this suggests that Sec may be actively keeping the stretch of pre-protein in the cytosolic cavity unfolded.

### Quantification of cross-translocon pre-protein secondary structure

To quantify the pre-protein folding more precisely, we carried out a series of additional simulations. Analysis of the initial trajectories (*Figure 2—figure supplement 2B*) suggests that a time scale of 110 ns is sufficient to kinetically sample helix formation. This matches time scales previously reported for model protein secondary structure formation (*Davis et al., 2015*), although it should be noted that any potential slower folding events, in the μs range, will not be sampled. We set up 36 independent simulations of ~110 ns (about 4 μs in total) per nucleotide state, varying both lipid composition and pre-protein sequence orientation (see Materials and methods for full details).

The simulations for each nucleotide state were analysed for hydrogen-bonded secondary structure in the 18 residue stretch of pre-protein through SecY. In the ATP-bound state there is >10 fold

**Table 1.** Details of remodelling performed on the crystal coordinates (*Li et al., 2016*).
In the left column is the residue number, as per the input model. In the second column is the residue sequence associated with that region of protein. Next is the name used here to describe this region. Names match the regions shown in *Figure 1—figure supplement 1B*. The last column briefly outlines how the region was modelled. See the text for more detail.

| Residues | Sequence | Name | Strategy |
|---|---|---|---|
| 246–249 | AEKD | SecA loop1 | Built using Modeller |
| 489–490 | RG | SecA loop2 | Built using Modeller |
| 620–624 | SENL | SecA loop3 | Built using Modeller |
| 643–683 | TPREELPEEWKLDGLVDLINT TYLDEGALEKSDIFGKEPDE | HWD 1 | Built from region in 3DIN |
| 700–705 | EEQFGK | HWD 2 | Built from region in 3DIN |
| 744–790 | | Substrate | Removed from SecA, made into new chain and extended through SecA by 30 residues |
| 738–748 | GGSGG | 2HF | Added new region for end of 2HF, based on 3DIN |
| 792–794 | QTN | SecA loop4 | Built using Modeller |
| 144–145 | GI | SecY loop1 | Built using Modeller |
| 207–213 | QTFGGLN | SecY loop2 | Modelled loop with Modeller based on Uniprot entry: QQFENVGEDLFLR |
| 245–258 | YAKRLEGRNPVGGH | C4 loop | Functionally important loop. Modelling based on 3DIN, including a short linker from *Thermotoga maritima* RITIQ to maintain 3D geometry. |
| 268–272 | PAGVI | SecY loop3 | Built using Modeller, and sequence shortened to fit space |
| 296–300 | DVTLWI | SecY loop4 | Built using Modeller |

more pre-protein structure apparent in the exterior cavity compared to the cytoplasmic cavity (*Figure 2C*; blue data). In contrast, in the ADP-bound state no significant asymmetry with respect to pre-protein secondary structure was observed (*Figure 2C*; red data). This suggests an ATP-dependent and highly asymmetric influence of the translocon on the folded state of the pre-protein; set up to permit folding on the exterior side of the channel only.

## Pre-protein folding asymmetry is enforced by the translocon

Further simulations were conducted to determine if this observed asymmetry was caused by the translocon, or was an intrinsic property of the chosen polypeptide sequence. The coordinates for the unfolded 18 residue stretch were extracted from the simulation and built into separate water boxes, without either SecA-SecYE or the membrane. Mild positional restraints were added to the ends of the peptide to replicate the constraints applied by the translocon and the bound signal sequence. These systems were then treated in the same way as the SecA-SecYE-PP simulations (i.e. simulated for 110 ns and each half analysed for secondary structure). The data show a small difference between the ADP and ATP states, likely a result of the mild positional restraints on the ends of the polypeptide, added to keep the substrate in a relatively transport-like conformation. This does not affect the primary outcome from this experiment, which is that no significant asymmetry in pre-protein formation is observed in either the simulation sets, be they derived from the ADP or ATP simulations (*Figure 2D*).

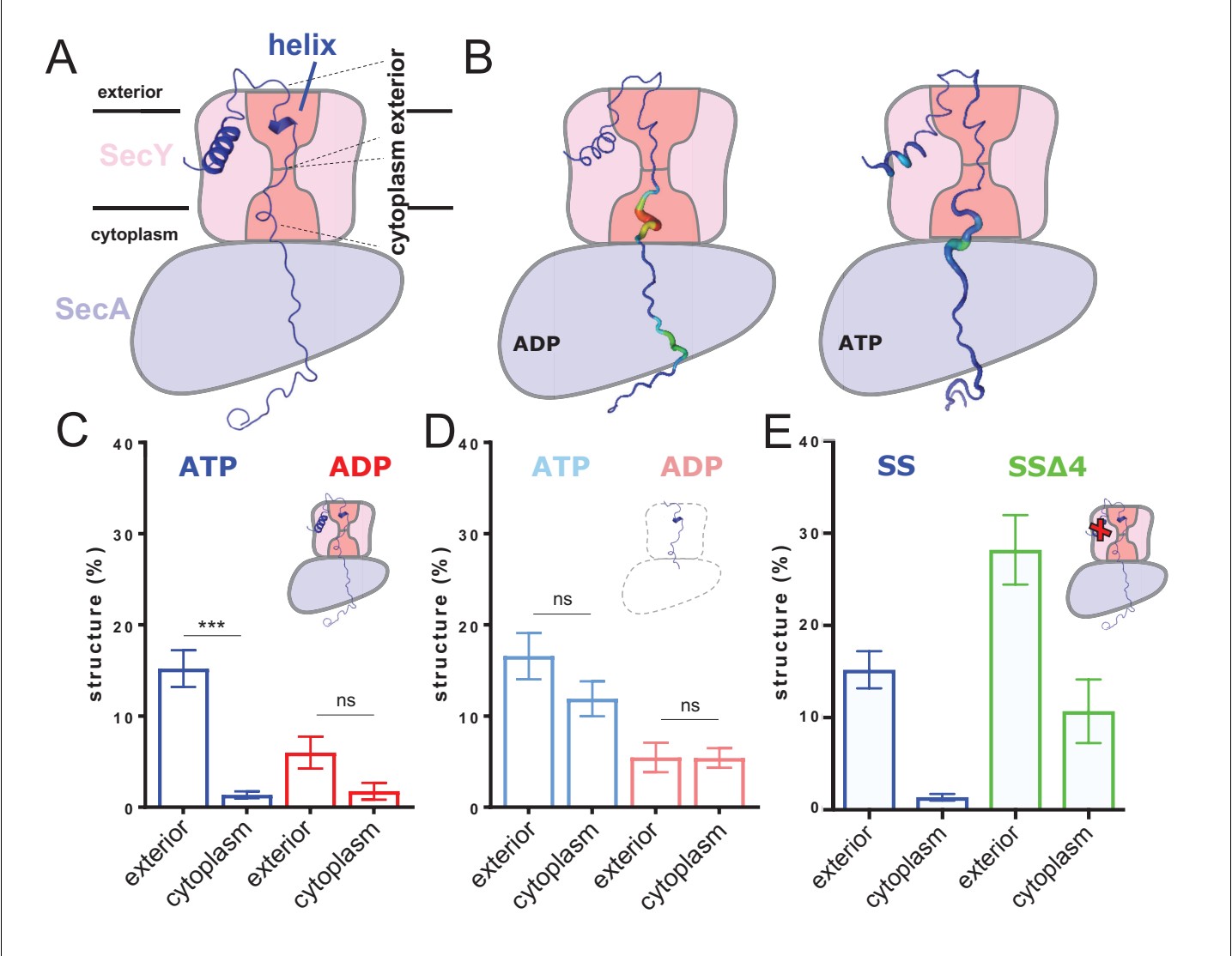

**Figure 2.** ATP-dependent asymmetric folding of pre-protein in the Sec translocon. (**A**) A 1 μs MD snapshot of the pre-protein from the SecA-SecYE-PP complex with ATP bound, overlaid on the SecA-SecY schematic from *Figure 1C*. A region of α-helix, as computed by the dictionary of secondary structure of proteins (DSSP) and confirmed visually, is visible in the exterior cavity. The approximate position of the membrane is shown, and each nine residue stretch of pre-protein used for folding analyses, respectively in the exterior and cytoplasmic cavities, are indicated by dashed lines. (**B**) Deformation analysis of the pre-protein within SecA-SecYEG. SecA-SecY is shown as a schematic, and the pre-protein is shown as tube, color-coded and sized according to its deformation energy (low deformation blue and thin; high deformation red and wide). Analysis reveals three major sites with high deformation energy; at the beginning of the signal sequence, in the SecY cytoplasmic cavity, and in the SecA ring. The former is more pronounced in the ATP state, the latter two in the ADP-bound state (see *Figure 2—figure supplement 4A*). (**C**) The degree of pre-protein folding in the exterior and cytoplasmic cavities of SecY in the ADP and ATP states. Shown are the combined datasets for the mirrored and tandem substrates (see Materials and methods for details), in either a simple or more complex bilayer supplemented with negative phospholipids (PG and cardiolipin). Data are collected from the ATP (blue) or ADP (red) bound states. There is a marked difference in degree of asymmetry in the ATP data, but not the ADP (p<0.0001 and p=0.0690 respectively, from two tailed *t*-tests). Error bars are s.e.m. The breakdown of data between uniform bilayers and those supplemented with negative phospholipids can be seen in *Figure 2—figure supplement 4E*. (**D**) As panel C, but showing the degree of folding in bulk water; that is, not in the presence of translocon. These analyses used the same pre-protein starting coodinates as panel C. In both the ATP (light blue) and ADP (pink) data, there is no significant difference between the cytoplasmic and exterior regions of pre-protein. Reported p values are 0.15 and 0.98. Error bars are s.e.m. (**E**) Comparison of folding data set with the wild-type signal sequence (blue; as per panel C) and a defective signal sequence (SS$_{\Delta4}$; green). The introduction of a defective signal sequence significantly increases the degree of pre-protein secondary structure in both the cytoplasmic and exterior cavities (p=0.021 and p=0.002). The breakdown of data can be seen in *Figure 2—figure supplement 4B*.

The online version of this article includes the following figure supplement(s) for figure 2:

**Figure supplement 1.** Time-evolving root-mean-squared displacements (RMSD) from the input coordinates.

*Figure 2 continued on next page*

**Figure supplement 2.** Structural highlights from the simulation data.
**Figure supplement 3.** Pre-protein structural changes in the channel.
**Figure supplement 4.** Expanded pre-protein folding data.

Therefore, the asymmetric secondary structure formation must be a consequence of the ATP-bound translocon, rather than the intrinsic properties of the pre-protein *per se*.

## Unfolding of the pre-protein requires the correct signal sequence

Next, we conducted a critical control experiment, by analysing the dependence of asymmetric pre-protein structure on the interaction with a functional signal sequence; required to 'unlock' the complex (*Corey et al., 2016*). For this, we constructed a system in which the engaged pre-protein possesses a defective signal sequence ('SS$_{\Delta 4}$'; missing residues 5–9), but retains its strategic position in contact with the lipid bilayer (*Hizlan et al., 2012*; *Briggs et al., 1986*; *McKnight et al., 1991*). Analysis of this complex reveals that the ATP-induced cross-channel asymmetry is somewhat reduced (*Figure 2C* and *Figure 2—figure supplement 4B*). However, more interestingly the degree of pre-protein secondary structure is significantly higher in both cavities; particularly in the cytosolic cavity (*Figure 2E*). Thus, a productive interaction of the signal sequence with the ATP-associated translocon seems to be required to reduce pre-protein folding in the cytosolic cavity. This suggests that the ability of the translocon to asymmetrically influence the folding propensity of the translocating pre-protein is subject to both activation by ATP and the signal sequence.

## No apparent role for specific protein-lipid interactions

Given the known dependency of protein secretion on anionic phospholipids, particularly cardiolipin (*Gold et al., 2010*; *Corey et al., 2018*; *Hendrick and Wickner, 1991*), we decided to look at their effects on the observed translocon-induced asymmetry. The data presented in *Figure 2C* combine simulations with different lipid compositions, including data in the presence of physiological concentrations of phosphatidylglycerol (PG) lipid and cardiolipin. For these, coarse-grained simulations were run on the post-1 μs ATP and ADP snapshots using the Martini force field (*Monticelli et al., 2008*; *Marrink et al., 2007*). Following 1 μs of simulation, multiple protein-lipid specific interactions were identified (*Figure 2—figure supplement 4C–D*), as per previous data (*Corey et al., 2018*). These systems were then converted back to an atomistic description (*Stansfeld and Sansom, 2011*) for further analyses.

Comparison of pre-protein folding in the presence and absence of PG/cardiolipin reveals no significant difference for any of the cavities (*Figure 2—figure supplement 4E*). Therefore, there appears to be no role for specific protein-lipid interactions in generating a cross-membrane pre-protein folding asymmetry.

## Perturbed translational dynamics of water molecules within the SecY translocon

The thermodynamics and kinetics of protein folding can be affected by perturbed water dynamics (*Lucent et al., 2007*), and water molecules within SecY have previously been shown to exhibit reduced mobility (*Capponi et al., 2015*). Therefore, to examine the role of water in the pre-protein folding process, 31 structural snapshots were extracted from each of the 1 μs SecA-SecYE-PP ATP and ADP simulations. Short simulations were run to model the water dynamics accurately (see Materials and methods for details).

To measure the translational water dynamics, mean squared displacement (MSD) calculations were employed (as per *Capponi et al., 2015*; see Materials and methods for details). When applied to the water molecules through SecA-SecYE-PP, there is a clear pattern of perturbation throughout SecY, with the water dynamics at the centre of the channel severely restricted (*Figure 3* and *Figure 3—figure supplement 1*). Comparison of the water molecules on either side of the pore reveal an asymmetry in the ATP-bound system – that is a higher degree of translational diffusion (disorder) in the cytoplasmic cavity than the exterior cavity – but not in the ADP-bound system.

Additional calculations were run for the translocon in a resting state, without pre-protein (*Figure 3—figure supplement 2*), which resembles the SecA-SecYE-PP in the ADP state, with no asymmetry between cavities. As with the analysis of pre-protein secondary structure formation, these data mark the cytoplasmic cavity of the ATP and pre-protein activated translocon as unusual in its environment. This observation suggests that the arrangement and ordering of waters within the SecY translocon may be contributing to the formation of pre-protein structure, with this effect – and hence the degree of pre-protein structure – lessened in the cytoplasmic cavity when bound to ATP.

However, it must be noted that it is unclear as to whether the perturbed water dynamics are affecting pre-protein folding/unfolding, or are a consequence of the folded state present.

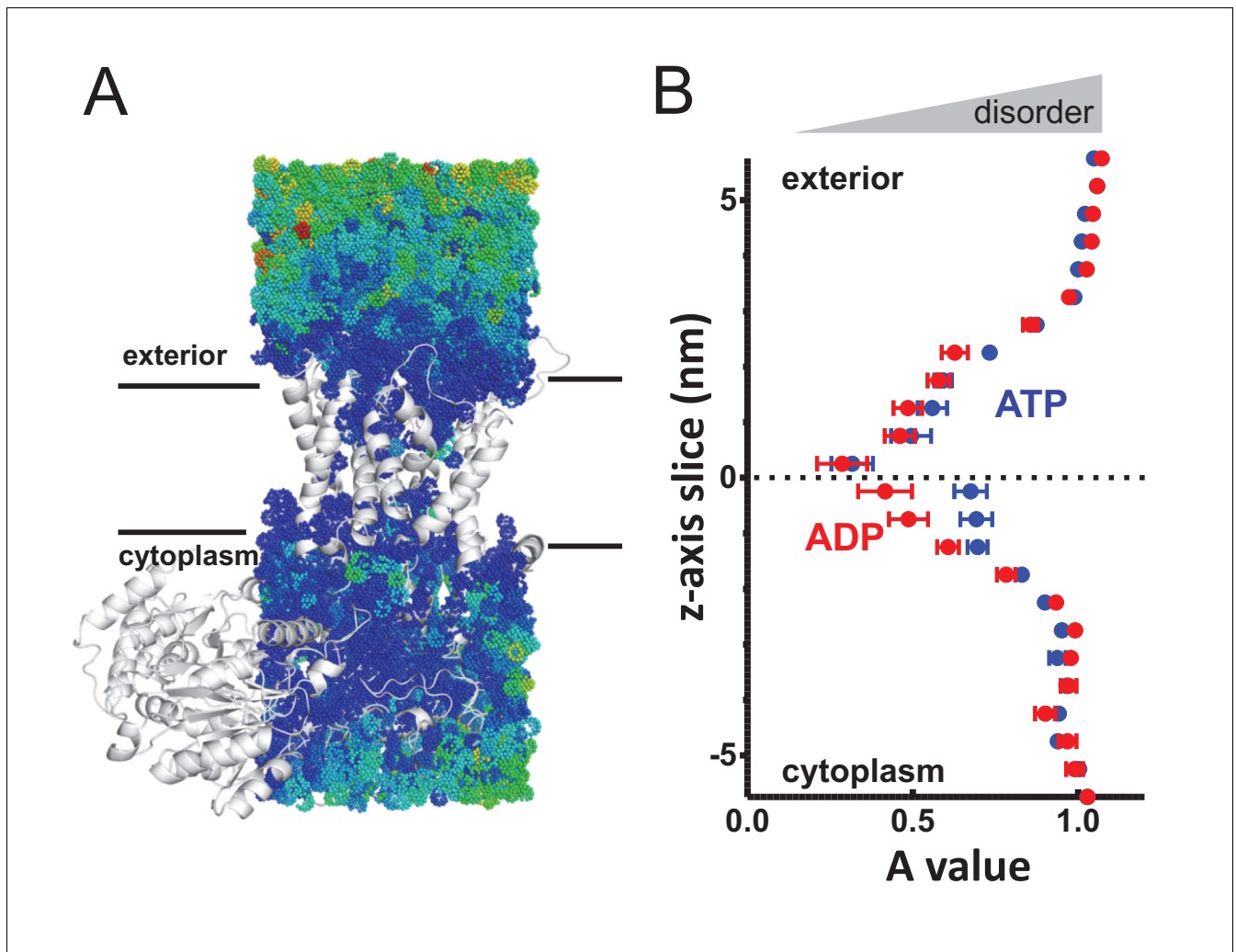

**Figure 3.** Perturbed water dynamics throughout the Sec translocon. (**A**) Cartoon showing the prism of waters analysed for translational dynamics. The waters are shown as coloured dots, coloured according to degree of diffusion, with blue slowest and red highest. Note that whilst the membrane and solvent outside of this prism are missing from this figure, they were present in the simulation. (**B**) MSD data of the waters along the length of the protein. The average MSD was calculated for each 0.5 nm horizontal slice, the data fitted to a power law equation, and the power value ('A') for each slice was averaged across all 31 simulations. Here, the average for each slice is shown, with s.e.m as error bars. Both the ATP and ADP simulations are perturbed in the centre of SecY, but the waters in the ATP-bound complex are perturbed in an asymmetric manner.

The online version of this article includes the following figure supplement(s) for figure 3:

**Figure supplement 1.** Raw MSD data from *Figure 3*.

**Figure supplement 2.** Analysis of waters through SecYEβ.

## Nucleotide-dependent variation in the geometry of the exterior and interior cavities of the translocon

The conformational entropy of an extended polypeptide is reduced when confined, which should favour more compact, folded states (*Zhou and Dill, 2001*). This raises the possibility that the ATP-driven opening of the SecY translocon (*Allen et al., 2016*) might have a direct impact on pre-protein secondary structure, whereby opening of the SecY cavities could promote pre-protein unfolding.

To test this, the cavity sizes of the translocon associated with either ATP or ADP were assessed throughout the 1 μs simulations. The pre-protein was removed from the translocon and the dimensions of the cavities on either side were measured for 31 structural snapshots from each simulation using the HOLE program (*Smart et al., 1996*). For consistency, the analyses were initiated at a set point between the residues Ile-78 and Ile-275 at the centre of the SecY pore (see *Figure 1—figure supplement 1A*). The cavity volumes were measured for 6.5 Å on either side of the pore, towards the cytosol (inside) or exterior (*Figure 4A*; *Figure 4—figure supplement 1A*). The relative sizes of these defined regions reveal an asymmetry between cytoplasmic and exterior cavities in the ATP-bound simulations (*Figure 4B*; blue data), but not in the ADP-bound simulations (*Figure 4B*; red data). The ATP cytoplasmic cavity is the outlier, being at least 10% larger than any of the others. This in turn reduces the degree of pre-protein contact with SecY (*Figure 4C*), suggesting a role in SecY-pre-protein contact in regulating secondary structure formation.

The pre-protein folding analyses (above; *Figure 2*) do not reveal the structural basis for asymmetric control of pre-protein folding; specifically, is it a consequence of promoted folding at the exterior, or prevention at the cytosolic cavity? The geometric analyses here suggest it is the latter, caused by a widening of the cytoplasmic cavity in the ATP-bound state.

## Cavity size regulation occurs in the absence of pre-protein

To test whether the ATP-driven increase in size of the cytosolic chamber (*Figure 4*) is dependent on pre-protein, we analysed previously produced simulation data (*Allen et al., 2016*) of the SecA-SecYEG complex (PDB code 3DIN [*Zimmer et al., 2008*]) with ATP or ADP bound, without pre-protein. Cavity size analyses reveal the same effect as described above: that is the cytoplasmic cavity of SecY is much larger than either the periplasmic cavity in the ATP-bound state or either cavity in the ADP-bound state (*Figure 4—figure supplement 1B–C*). Evidently, the cytoplasmic cavity opens up in response to ATP irrespective of the presence of pre-protein.

## Validation of MD simulations showing asymmetric SecY cavity size by EPR

Experimental measurement of localised and transient secondary structure changes within a highly dynamic Sec-engaged pre-protein is a challenging prospect, arguably beyond our current capabilities. Instead, we chose to experimentally analyse the observed asymmetry in SecY cavity sizes. To this end, we applied the EPR technique of double electron-electron resonance (DEER; otherwise known as PELDOR) spectroscopy, which allows detailed conformational sampling of the distances between two spin labels attached at specific positions to a molecule (*Milov et al., 1981*; *Martin et al., 1998*; *Jeschke, 2012*).

Firstly, we conducted MD simulations of the *T. thermophilus* SecYEG resting state structure (PDB 5AWW [*Tanaka et al., 2015*]) labelled with nitroxide-containing spin label 'MTS' (MTS; S-(1-oxyl-2,2,5,5-tetramethyl-2,5-dihydro-1H-pyrrol-3-yl)methyl methanethiosulfonate) (*Berliner et al., 1982*; *Figure 5—figure supplement 1A*) covalently attached to two cysteines on either sides of the SecY channel: residues Ile-58 and Ser-101 (*Figure 5—figure supplement 1B*; *Fessl et al., 2018*). The resultant simulation provides an ensemble of modelled DEER data for the labelled translocon based on the experimentally determined closed structure.

Next, we recreated the effect of SecA-ATP binding through targeted MD, using SecY from the SecA-SecYEG crystal structure (*Zimmer et al., 2008*) as a template, and with the targeting force constants being increased every 100 ps over 800 ps (*Figure 5—figure supplement 1C*). This allowed us to produce an ensemble of conformations of spin-labelled complexes based on the experimental open structure. Comparing the two ensembles, we observe that transitioning to the open structure reduces the inter-label distance substantially (*Figure 5A–C*). This is mainly caused by resizing of the central cavity, which permits one of the spin labels to flip inside (*Figure 5B*). Note that the cavity

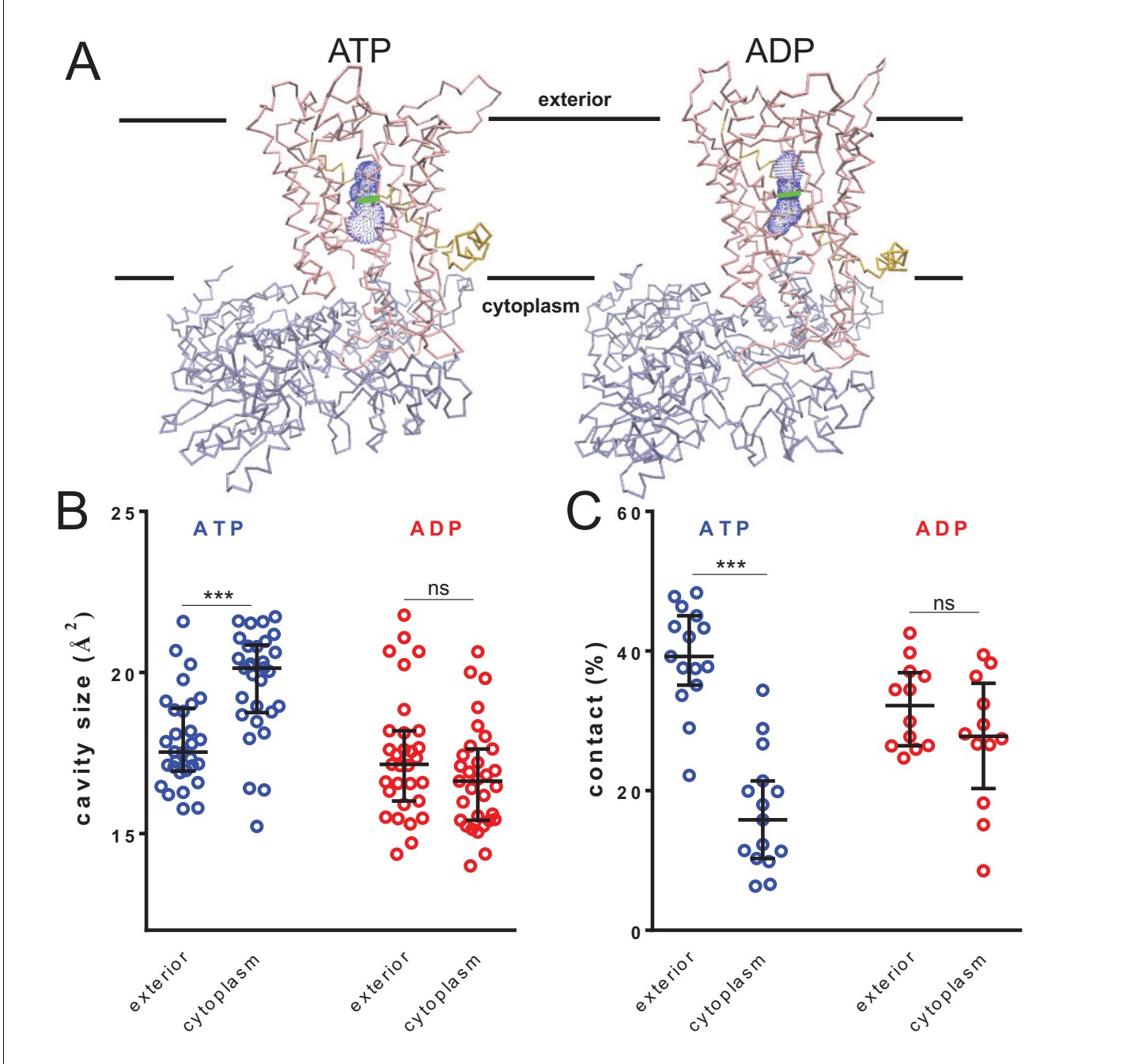

**Figure 4.** Nucleotide-driven cavity resizing in SecY. (**A**) Representative HOLE data used for averaging analysis. The protein backbone is shown as coloured trace and the calculated cavity as blue mesh. The central coordinate from the HOLE program is shown as a green line. (**B**) Quantified cavity sizes from the ATP and ADP simulation data, plotting the median and interquartile range. The difference between the exterior and cytoplasmic cavities was tested against an unpaired two-tailed $t$-test, reporting p values of < 0.0001 and 0.1264 for ATP and ADP, respectively. (**C**) Degree of contact between the pre-protein and SecY channel, defined as inter-residue distances of less than 0.3 nm, averaged over all of the residues. Correlated with a wider cavity, there is significantly less contact in the cytoplasmic cavity in the ATP-bound state (p<0.0001), but not in the ADP-bound stage (p=0.1404). The online version of this article includes the following figure supplement(s) for figure 4:

**Figure supplement 1.** Cavity size analyses.

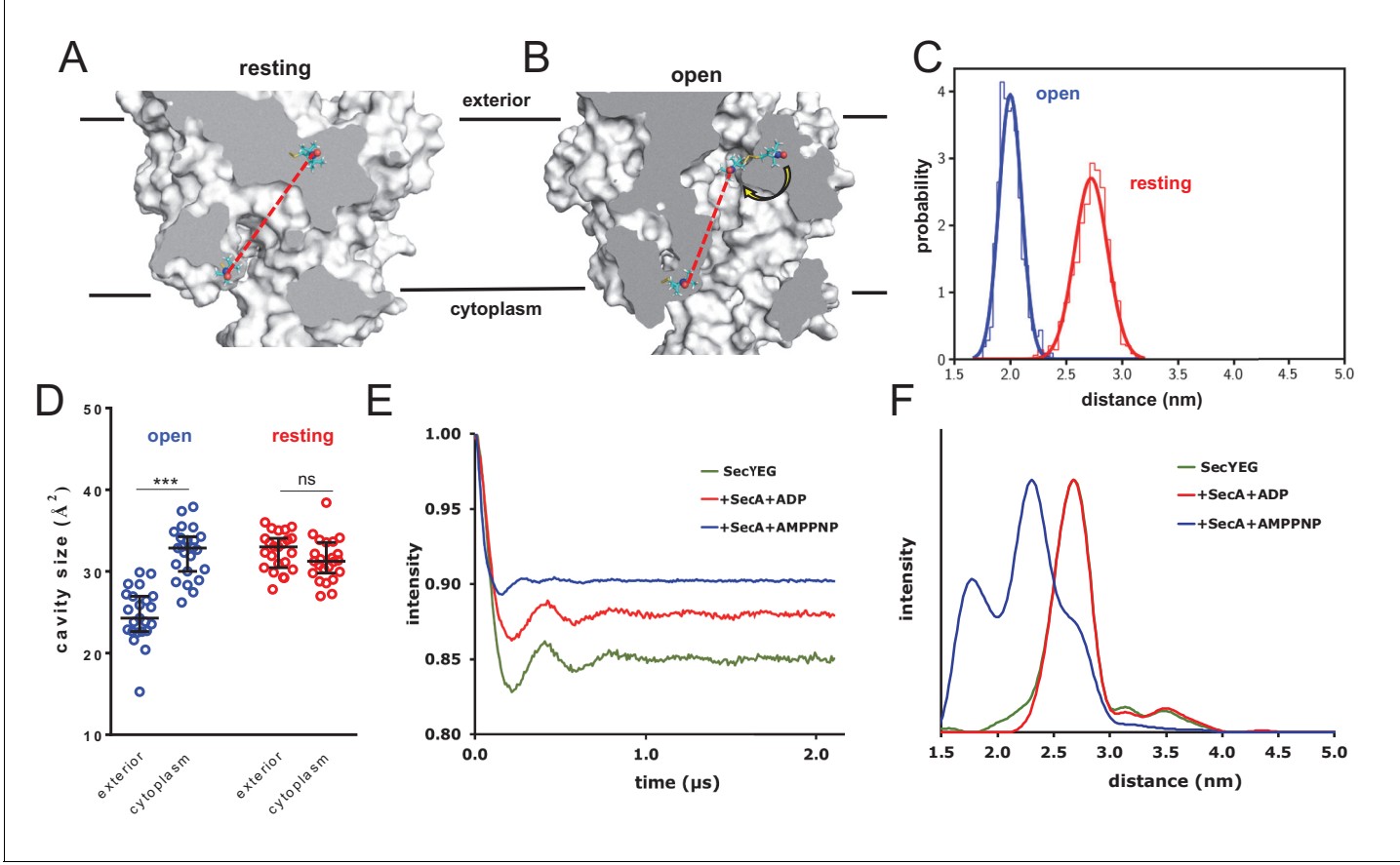

**Figure 5.** Combined EPR and MD supports nucleotide-driven cavity resizing. (**A**) Snapshot from the resting spin-labelled SecYEG simulation, showing the protein as grey surface and slabbed to show the interior of the channel. The spin label carbons are shown as cyan sticks, with the spectroscopically-relevant NO group in blue and red spheres. The disulphide bond to the SecY cysteine residues is shown in yellow. A predicted DEER distance is shown as a red line. (**B**) As panel A, but of the open channel. Upon channel widening, the MTS is able to flip into the channel (yellow arrow), resulting in a much shorter predicted DEER distance **C**) Histograms of intra-MTS distance data for the resting MD simulation (red) and the opened simulation (blue), overlaid with a normal distribution. The x-axis has been set as per panel F. (**D**) HOLE cavity analysis of the open and resting channel with MTS labels attached. The systems based on the open structure exhibit an asymmetry with respect to cavity size (blue data), whereas simulations based on the closed PDB exhibit no asymmetry (red). (**E**) Background corrected DEER time traces of spin-labelled SecYEG alone (green) and with excess SecA in the presence of either ADP (red) or AMPPNP (blue). The dipolar coupling between the nitroxide spin labels is evident in all three traces as modulations on the intensity of the detected spin-echo. This is almost the same for the spin-labelled SecYEG with SecA/ADP or without. (**F**) Distance distributions obtained from DeerAnalysis2016 (*Jeschke et al., 2006*) from the DEER time traces shown in E. The results for spin-labelled SecYEG with SecA and AMPPNP are, on average, at a shorter distance with a broader distribution than the overlapping distance distributions for spin-labelled SecYEG alone or with SecA/ADP. More information on the implementation of DeerAnalysis is given in the Supporting Information.

The online version of this article includes the following figure supplement(s) for figure 5:

**Figure supplement 1.** MD data in support of EPR DEER analysis.
**Figure supplement 2.** Raw EPR DEER data.

sizes are asymmetric in the open ensemble but not in the resting state (*Figure 5D*) exactly as for the other simulation sets (*Figure 4B*).

Experimental DEER spectra were then obtained by attaching spin labels at the equivalent sites on *E. coli* SecYEG, and measuring the inter-nitroxide distance (*Martin et al., 1998*; *Jeschke, 2012*; *Haugland et al., 2016*; *Jeschke et al., 2006*; *Todd et al., 1989*) for SecYEG alone; with SecA and ADP; and with SecA and the non-hydrolysable ATP analogue AMPPNP. The results reveal a strong nucleotide-dependent effect on channel conformation (*Figure 5E–F* and *Figure 5—figure supplement 2*). The most probable distance between the two sites in the SecA-SecYEG complex with ADP is ~2.7 nm, with a full width at half height (FWHH) of ~0.4 nm (*Figure 5F*), matching the MD data extremely well (*Figure 5C*). This value is identical to the distance for SecYEG alone – supporting

earlier findings that the resting SecY and ADP-bound SecA-SecYEG states are very similar with respect to the channel region (*Allen et al., 2016*). Addition of AMPPNP, meanwhile, causes the two positions to rearrange, with the spin labels coming much closer together to give a modal distance of ~2.3 nm and broader FWHH of ~0.9 nm (*Figure 5F*) – again, in good agreement with the MD data (*Figure 5C*). These experimental distances are only an indirect measure of cavity opening, however they do provide direct experimental validation for the MD data, and confirm that ATP does indeed have a potent effect on the conformation of the channel.

## HDX-MS reveals asymmetry in SecY cavities

To further explore the environment and conformational dynamics of the protein-channel during the ATPase cycle, we carried out differential HDX-MS experiments on *E. coli* SecA-SecYEG. HDX-MS reports on the exchange of hydrogen to deuterium in backbone amides of a protein or protein complex (*Englander and Kallenbach, 1983*; *Konermann et al., 2011*; *Engen, 2009*). The exchange rates are dependent on the protein solvent accessibility and hydrogen bonding (*Wales and Engen, 2006*). The main strength of HDX-MS is it allows non-invasive monitoring of the dynamics of an unmodified protein complex at peptide-level of resolution, enabling comparison between distinct protein states. Here, we directly compare the HDX rates between the ADP- and ATP-bound states, aiming to gain quantitative information on how nucleotide affects global conformational changes in the SecA-SecYEG complex, including networks within the SecY cavities.

We performed differential HDX-MS experiments on SecYEG saturated with SecA and either ADP or AMPPNP (hereafter referred to as ATP, for clarity). The nucleotide-bound conditions require SecA in a slight molar excess over SecY, so any data pertinent to SecA was ignored. By mapping the difference in deuterium uptake between the ATP- and ADP-bound states (ΔATP–ADP) on the crystal structure of SecA-SecYEG (PDB 3DIN), we showed significant changes primarily located in the periplasmic and cytoplasmic regions (*Figure 6A* and *Figure 6—figure supplement 1A–C*). Specific peptides were identified with significant ΔHDX (99% confidence interval), as useful reporters for changes in these two regions of interest (*Figure 6B* and *Figure 6—figure supplement 1A*).

Peptides lining the cytoplasmic cavity (CC) show deprotection induced by the presence of ATP (4 out of 4 peptides; *Figure 6B–C* and *Figure 6—figure supplement 1B*). This is consistent with a destabilisation of this region and may indicate a more 'open' cavity. In contrast, from the two peptides identified that line the exterior cavity (PC; not including the 'plug', see below), one of them revealed protection and the other a mild deprotection suggesting a more limited effect of ATP on the exterior cavity (*Figure 6B,D* and *Figure 6—figure supplement 1C*). This is consistent with the MD simulations which also showed mild dynamic changes in these regions.

Interestingly, detailed inspection of the equivalent regions in the MD simulation data reveals striking similarities with the HDX-MS results. For instance, a peptide in the cytosolic region between TMS 6–7 (CC/1), which lines the protein channel is subject to a massive increase in HDX with ATP (~50% compared to ~10% with ADP; *Figure 6C* and *Figure 6—figure supplement 1B*). This region is very highly conserved (*Figure 6C* and *Figure 6—figure supplement 1G*) and in the simulation data exhibits a substantial change in its structure between the ATP and ADP associated states (*Figure 6—figure supplement 1H*). The ATP-dependent mobilisation of this area does indeed appear to increase the size of the cavity and therefore could influence the folded state of the adjacent translocating pre-protein. Together, these analyses provide compelling evidence for a nucleotide-driven asymmetric resizing of the SecY cavities.

Overall the data confirm the presence of long-range conformational changes from the nucleotide binding site on SecA to the protein-channel (*Figure 1C*) (*Fessl et al., 2018*; *Allen et al., 2016*; *Robson et al., 2007*). Many of these differences occur in regions of SecYEG known to be susceptible to remote nucleotide modulation (*Figure 6—figure supplement 1D*). For example, the increased HDX (ATP *versus* ADP) observed in the 'plug' (extended plug loop; EPL) of SecY and the amphipathic helix (AH) of SecE are consistent with their known mobilisation in the translocon when activated by ATP (*Corey et al., 2016*). The contrasting decrease in HDX seen in the two largest cytosolic loops (CL) of SecY (*Figure 6—figure supplement 1D*) reflects the tighter association with SecA in the ATP bound state (*Robson et al., 2007*; *Robson et al., 2009*). Therefore, our HDX-MS experiments clearly support the notion of ATP-induced asymmetry in the SecYEG channel.

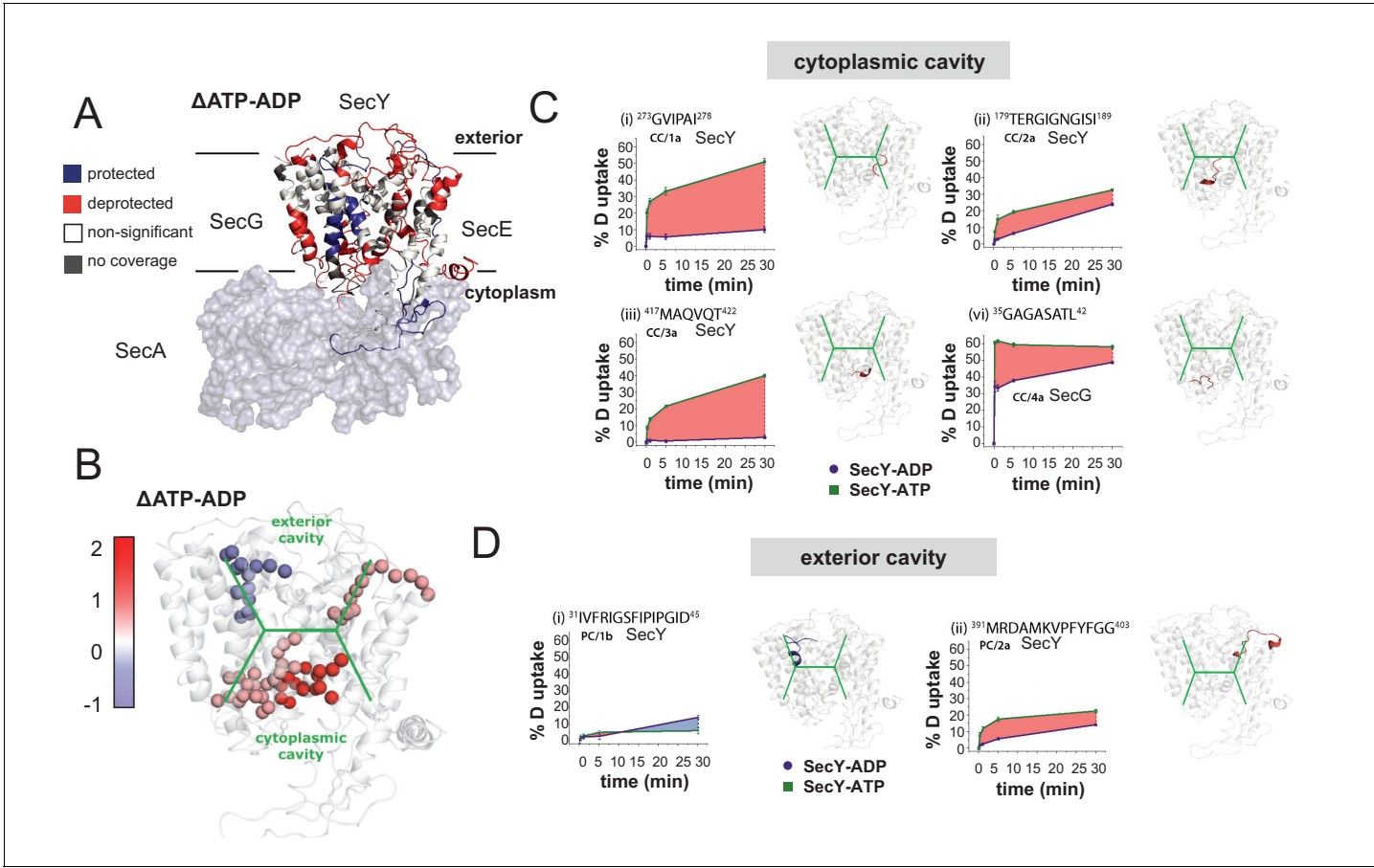

**Figure 6.** HDX data demonstrates ATP-driven cavity changes. (**A**) Differences in relative deuterium uptake (ΔHDX) (ΔATP-ADP) of SecYEG after a 30 min exposure to deuterated solvent. A significant change in ΔHDX was determined as >0.9 Da (99% CI). Blue and red coloured regions indicate peptides that become HDX protected or deprotected, respectively. White regions represent peptides where no significant ΔHDX is observed. Regions with no coverage obtained are coloured grey. SecA data have been removed for clarity. HDX data are mapped onto a SecA-SecYEG complex structure (PDB: 3DIN). (**B**) View of SecYEG from PDB 3DIN (*Zimmer et al., 2008*). Shown in coloured spheres are backbone nitrogens which are located in the SecY cavities, and which exhibit a significant difference in deuterium exchange between the ATP and ADP states. The colour represents the difference in magnitude (blue = higher exchange in the ADP state, red = higher exchange in the ATP state). The approximate positions of the respective cavities are shown in green. (**C–D**) Deuterium uptake plots of peptides in the SecYEG cytoplasmic cavity (CC, residues 273–278, 179–189 and 417–422 in SecY, and 35–42 in SecG) and the periplasmic cavity (PC, residues 31–45 and 391–403 in SecY). *E. coli* residue numbering used throughout. On the right of each panel, the peptides are highlighted on the equivalent position of the SecYEG crystal structure from PDB 3DIN (*Zimmer et al., 2008*).

The online version of this article includes the following figure supplement(s) for figure 6:

**Figure supplement 1.** Full HDX data.

## Pre-protein secondary structure prevents transit through SecY pore

To determine whether or not the pre-protein secondary structures observed in the simulation data impede passage through the SecY pore, a series of steered MD simulations were carried out. In these, a directional pulling force was applied to a stretch of the pre-protein through SecYE, with SecA and the rest of the pre-protein removed (*Figure 7A*). The pre-protein contained either an α-helical region – as sampled by equilibrium MD in the ATP- or ADP-bound states (*Figure 2*), and stabilised here using a distance restraint between positions *i* and *i + 4* – or had the helix abolished using a short steered MD simulation. These substrates were then pulled through the SecY pore, and the passage time of 10 independent repeats recorded. Note that the translocation of α-helical pre-protein is unlikely to occur in the physiological system. These simulations are designed not to determine time to pass a folded region through the SecY pore, rather to report on the ease with which different pre-protein structural states might freely diffuse through.

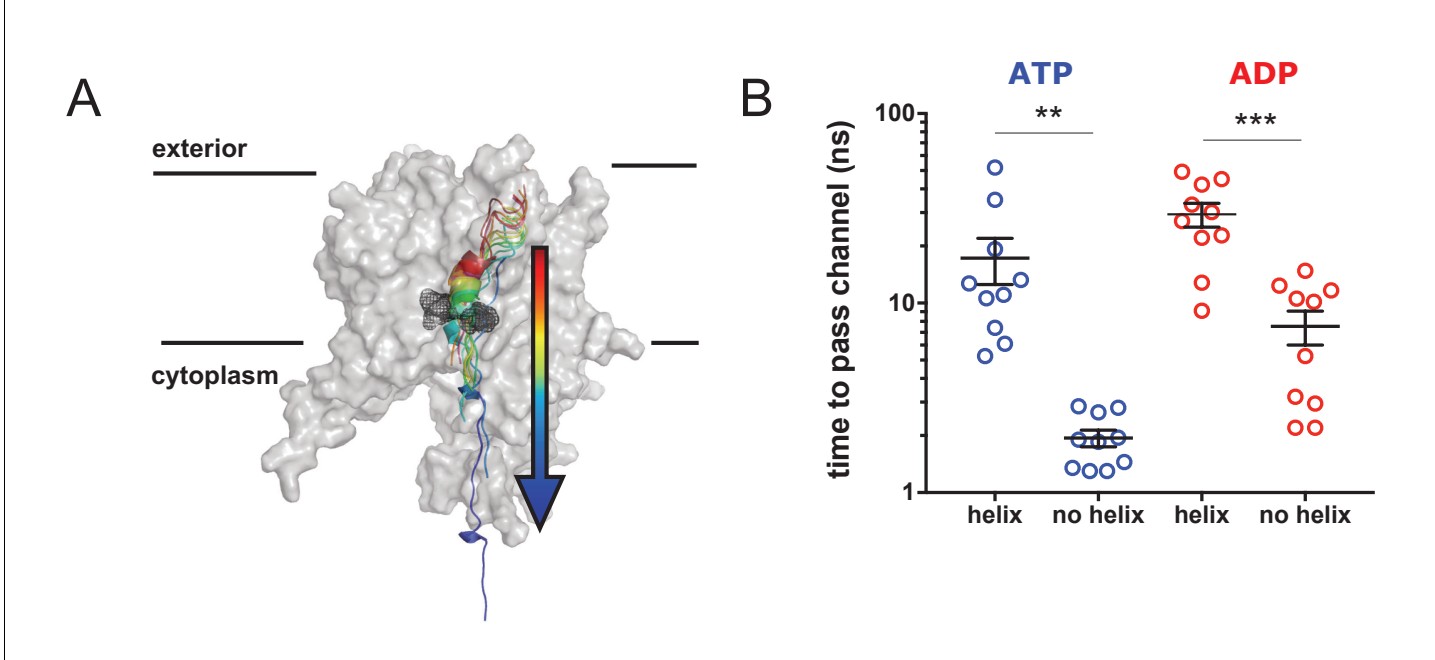

**Figure 7.** Pre-protein secondary structure prevents transit through SecY. (**A**) Cartoon illustrating the principle behind the steered MD simulations. From a 1 μs snapshot of ADP- or ATP-bound SecA-SecYE-PP, SecYE and a short region of helix-containing pre-protein were extracted. Two configurations of the pre-protein were established: either the helix was stabilised by a distance restraint between the hydrogen bonded atoms, or the helix was broken by a short steered MD simulation pulling the ends apart. A backward pulling force was then applied to each pre-protein, and the time taken to pass the channel measured (**B**) Steered MD data. The presence of a helix in the pre-protein strongly restricts passage though the pore, p=0.0045 and p=0.0001 using two-tailed t-tests.

The results show that the helix-containing substrate takes considerably longer to pass (~5–10 times slower) than the substrate with no helix (*Figure 7B*) in both the ATP- and ADP-bound states. Additionally, the ADP-bound state is generally more restrictive to transport than the ATP-bound state, supporting the concept of nucleotide regulated pore opening and closure (*Allen et al., 2016*). Although this setup clearly does not represent the physiological translocation process, the data indicate a significant barrier to the transit of folded regions of pre-protein through the protein channel.

## Discussion

While it is widely accepted that pre-proteins need to be unfolded during transport (*Arkowitz et al., 1993*; *Bonardi et al., 2011*), it is unclear as to the degree of this unfolding. The general assumption is that the tertiary structure forms after the translocation process, but what of the secondary structure? Does it form prior, during or after protein transport? This study seeks to investigate this problem and explore how the folding process *per se* might be exploited in the mechanism of protein transport.

A high degree of secondary structure has been reported for extended pre-proteins as they exit the ribosome (*Lu and Deutsch, 2005*; *Hardesty and Kramer, 2001*), and many proteins fold further once they emerge into the periplasm (*Antonoaea et al., 2008*). Furthermore, previous studies have intimated that pre-protein can form secondary structure within the translocon (*Gumbart et al., 2011*; *Zhang et al., 2017*). Here, we have combined computational and structural experiments on the SecA-SecY complex, aimed at elucidating the importance of pre-protein folding for the process of protein translocation.

The data shown here reveal a propensity for the pre-protein to form secondary structure in both the cytoplasmic and exterior cavities of the channel, when associated with SecA bound to ADP. Strikingly, when ATP is bound to SecA, this propensity becomes highly asymmetric, with secondary structure strongly disfavoured in the cytoplasmic cavity (*Figure 2*). In this state, the cavity also

exhibits a marked change in environment and increased size (*Figures 3B* and *4B*), confirmed by EPR DEER spectroscopy (*Figure 5*) and HDX MS (*Figure 6*). This also causes a decrease in contact between the translocon and pre-protein (*Figure 4C*). These observations suggest a specific role for ATP (and its hydrolysis to ADP) in regulating the structure of the translocating pre-protein: ATP binding causes the SecY cytoplasmic cavity to expand, reducing the amount of secondary structure, relative to the amount formed in the exterior cavity.

Steered MD simulations confirm that regions of secondary structure are much less likely to pass the narrow SecY pore (*Figure 7*). Taken together, these effects could contribute significantly to any translocation model that incorporates pre-protein diffusion within the channel, including full diffusion-based models (*Allen et al., 2016*) and hybrid diffusion/power stroke models (*Bauer et al., 2014*). Interestingly, the data also argues against full power-stroke based models (*Erlandson et al., 2008a*), as the pre-protein conformational changes observed here are apparently unrelated to motions of the 2HF.

The results described here allow us to extend our previously proposed 'Brownian ratchet' model for protein translocation (*Allen et al., 2016*), outlined in *Figure 8*. The initiation process involves ATP dependent activation of SecYEG by SecA and the pre-protein signal sequence (*Fessl et al., 2018*; *Corey et al., 2016*; *Hizlan et al., 2012*; *Gold et al., 2013*), and results in an intercalated diffusible mature region of pre-protein (*Figure 8*). When the translocon is bound to ADP, unstructured polypeptide in this region can flow freely within the channel, which explains why 'backsliding' can occur in the ADP bound state (*Erlandson et al., 2008b*). Meanwhile, secondary structure can equally form in the exterior or cytosolic cavities, producing no bias for forward or backward translocation. However, the appearance of secondary structure in the cytosolic cavity promotes nucleotide exchange, ADP for ATP (*Allen et al., 2016*) (*Figure 8*, bottom left). As we show in this study, the ATP bound translocon strongly favours secondary structure formation in the exterior cavity, but not so in the cytosolic cavity. Given that the folded domains cannot backtrack, this favours the forward diffusion of polypeptide (*Figure 8*, upper middle). Successive rounds of hydrolysis and exchange continue until transport is complete and the signal sequence is removed by proteolysis (*Figure 8*, centre right). Since, during translocation the steady-state complex is dominated by the ATP-bound state (*Robson et al., 2009*) (*Figure 8*, green box), the observed asymmetry is maintained throughout (upper states predominate); thus preventing backsliding and favouring productive transport.

This new mechanism for forward transport through the channel, along with additional stimulation from the trans-membrane PMF (*Brundage et al., 1990*), could achieve the required very fast rates required for efficient protein secretion. This mechanism would also explain observed differences in the rates of transport (*Fessl et al., 2018*; *Liang et al., 2009*; *Tomkiewicz et al., 2006*) and stoichiometry of ATP: pre-protein (*Tomkiewicz et al., 2006*; *van der Wolk et al., 1997*), because they would be dependent on the variable propensities of different protein sequences to form secondary structures. Interestingly, it could also explain the higher translocation efficiency of the *prl* mutants (*Bieker et al., 1990*), which have been shown to exhibit a more open conformation (*Corey et al., 2016*), thus favouring the ATP-bound conformation.

Presently, it is only possible to simulate time scales (μs) much shorter than the physiological rates of translocation (40 aa sec$^{-1}$ [*Fessl et al., 2018*]). Nevertheless, the kinetically fast event we sample – the formation of pre-protein structure – may well be critical for the transport mechanism.

It should be noted that the identified secondary structures might not reflect the final structure of the transported protein, especially for the β-barrel proteins of the outer membrane. However, they may correspond to folding intermediates proposed to be important for translocation competence through the Sec machinery (*Tsirigotaki et al., 2018*). The reliance on protein folding to aid the ratcheting process might predict that an inherently disordered protein substrate would fail to transport, as indeed it does in all of the bacterial and eukaryotic counterparts (Sec61; both yeast and mammalian [*Gonsberg et al., 2017*]). In cases where the propensity for folding is too great, such that it cannot be prevented as described, then other mechanisms could prevail. The *E. coli* thioredoxin DsbA does just that and is transported across the membrane co-translationally (*Huber et al., 2005*). Other fast folding proteins could be selected for secretion through the TAT machinery capably of transporting unfolded proteins (*Berks, 2015*).

Given the universal nature of secondary structure in polypeptides, such conformational control could be central to many other protein translocating machines, including chaperonins, organellar protein importers and the secretion systems of bacterial pathogens.

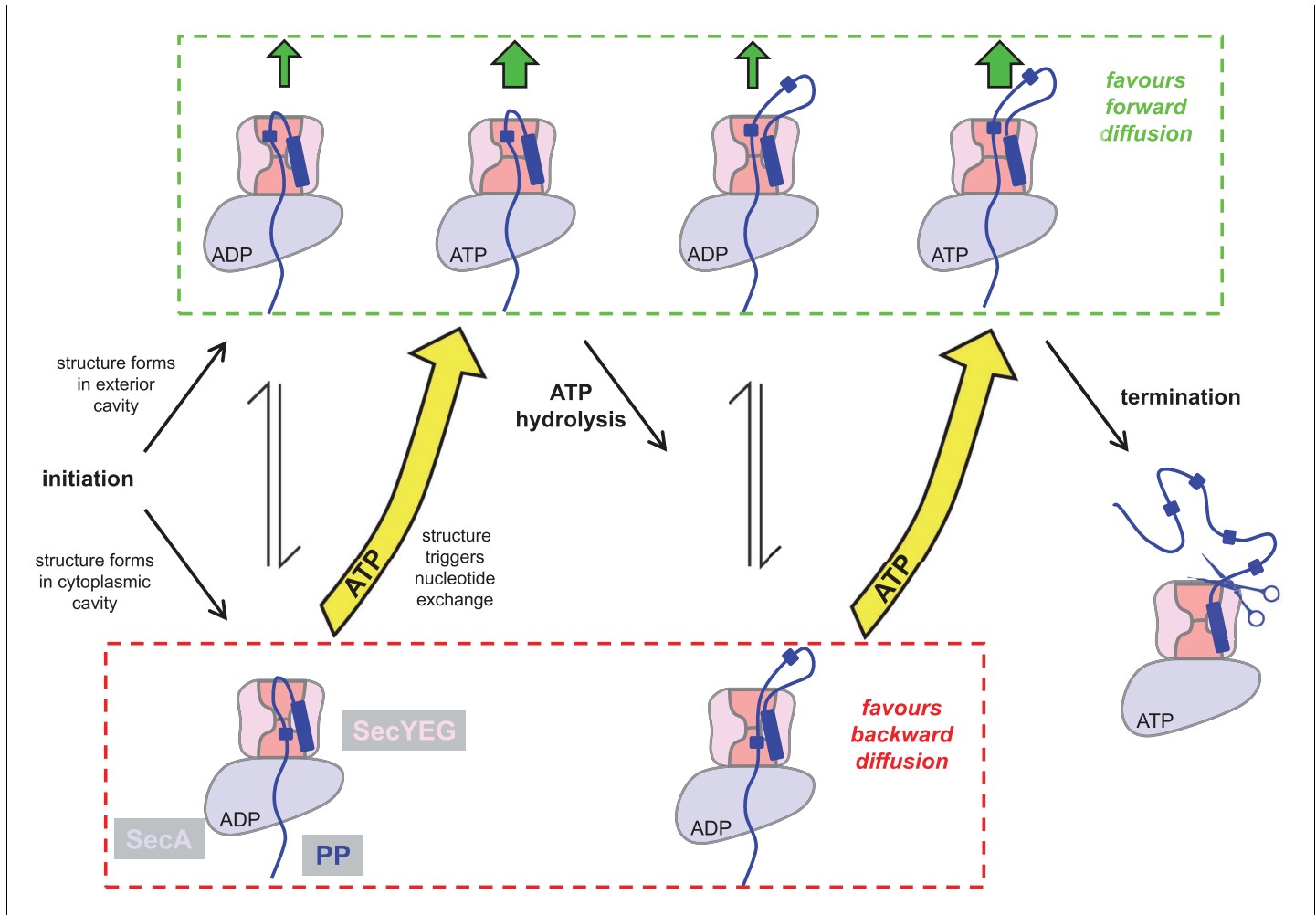

**Figure 8.** Extended model for pre-protein translocation through SecA-SecYEG. In this model, SecA is shown in light blue, SecYEG in light pink, the SecY central cavity in dark pink and the pre-protein in dark blue, as per *Figure 1*. Following initiation (centre left), localised pre-protein secondary structure formation can occur in either the exterior or cytoplasmic cavity, with roughly equal chance between compartments in the ADP state. Structure in the exterior cavity favours forward – productive – diffusion (top left state), because the structured region cannot diffuse backwards. Structure in the cytoplasmic cavity would favour backward – unproductive – diffusion (bottom left state). We previously demonstrated that bulky pre-protein in the cytoplasmic cavity triggers nucleotide exchange (*Allen et al., 2016*). Here, we demonstrate that ATP induces a strong asymmetry across the channel: secondary structure is much more likely to form in the exterior cavity (upper middle state). Thus, in the ATP-bound state productive forwards diffusion is favoured over backwards. At this point, either translocation proceeds to completion, or ATP hydrolysis resets the complex for successive cycles of ATP hydrolysis and ADP exchange for ATP (right). In this way, ATP is acting to prevent the backward diffusion of pre-protein in the complex, by shifting the equilibrium away from the bottom (unproductive) states (yellow arrows).

# Materials and methods

### Modelling the SecA-SecYEG-PP complex

The bulk of the data here are collected from a SecA-SecYEG-PP complex in an ATP or ADP state. Models for the simulations were built using chains A, Y and E of the crystal structure 5EUL (*Li et al., 2016*) as starting coordinates. Several small missing loops were added using the Modeller program (*Sali and Blundell, 1993*), detailed in *Table 1*. Some larger loops were added manually based on similar structures of SecA-SecYEG (PDB code 3DIN; *Zimmer et al., 2008*). These include HWD 1 and 2, two periplasmic SecY loops and a cytoplasmic SecY loop. The details of these loops are given in *Table 1*, and the loops are shown in *Figure 1—figure supplement 1B*. We reasoned that this would produce a better model than loops built simply based from sequence.

The original structure has a non-physiological arrangement of the SecA 2HF and bound pre-protein substrate, where these are represented by a single, continuous chain. To correct for this, we remodelled the tip of the 2HF based on 3DIN, to generate a physiological 2HF conformation, which is stable during simulation (*Figure 2—figure supplement 1*). In the SecA NBS, the ADP-BeFx molecule was replaced with either ADP or ATP (*Piggot et al., 2012*). The substrate was extended in an unfolded conformation through the SecA ring, *via* known crosslinking sites (*Bauer and Rapoport, 2009*). Substrate building was done with PyMOL (*Delano, 2002*).

## Molecular dynamics simulations

All simulations were run using GROMACS 4.6.4 or 5.1.2 (*Berendsen et al., 1995*). Simulations were built using the SecA-SecYEG-PP complex described above, or alternatively the *M. jannaschii* SecYEβ crystal structure (*Van den Berg et al., 2004*) or the *T. thermophilus* SecYEG crystal structure (*Tanaka et al., 2015*).

In most cases, the protein and solvent atoms were described in the OPLS all-atom force field (*Jorgensen et al., 1996*), with the simulations being run in an OPLS united-atom POPC membrane (*Ulmschneider and Ulmschneider, 2009*). The protein-membrane structures were built into simulation boxes with periodic boundary conditions in all dimensions and solvated with explicit SPC water and sodium and chloride ions to a neutral charge and concentration of 0.15 M. For the 1 μs SecA-SecYE-PP and *M. jannaschii* SecYEβ simulations, the systems were energy minimized using the steepest descents method over $2 \times 5000$ steps, then equilibrated with positional restraints on heavy atoms for 1 ns in the NPT ensemble at 300 K with the Bussi-Donadio-Parrinello thermostat (*Bussi et al., 2007*) and semi-isotropic Parrinello-Rahman pressure coupling (*Parrinello and Rahman, 1981*; *Nosé and Klein, 1983*).

Production simulations were run without positional restraints with 2 fs time steps over 500–1 μs on the UK HPC facility ARCHER, using time awarded by HECBioSim. Bond lengths were constrained using the LINCS method. Non-bonded interaction cut-offs were calculated using the Verlet method, with the neighbour search list updated every 20 steps. Long-range electrostatic interactions were calculated using the particle mesh Ewald method and a cut-off of 1.0 nm was applied for van der Waals and short range electrostatic interactions. All simulations reached a steady state as judged by their root-mean-squared-deviation from the starting structure (*Figure 2—figure supplement 1*).

For some of the folding data, membranes consisting of 63% POPE, 32% POPG and 5% cardiolipin in a coarse-grained description were built around the protein using the MemProtMD program (*Stansfeld et al., 2015*). Following 1 μs simulation, snapshots were obtained in which multiple acidic lipids were seen to bind to the complex (*Figure 2—figure supplement 4C–D*). These were then converted to an atomistic description (*Stansfeld and Sansom, 2011*), and the systems simulated using the Charmm36 force field (*Best et al., 2012*).

## Deformation analysis on pre-protein

Deformation energies were calculated from the SecA-SecYEG-PP ADP and ATP MD trajectories. Deformation analysis based on normal mode (vibrational) analysis provides a measure for the amount of local flexibility in the protein structure - that is atomic motion relative to neighbouring atoms. We calculated deformation energies and atomic fluctuations of the first three non-trivial modes in Bio3D (*Hinsen, 1998*; *Grant et al., 2006*) and visualized the results in PyMOL (*Delano, 2002*).

## Targeted MD

For the systems built from *T. thermophilus* SecYEG (*Tanaka et al., 2015*), simulations were initially run of the resting state of the channel (i.e. SecYEG alone), with MTS spin labels attached to cysteine residues engineered at positions 58 and 101. MTS parameters were produced using ACPYPE (*Sousa da Silva and Vranken, 2012*). MD simulation of the MTS in water produces the appropriate geometries (*Figure 5—figure supplement 1A*).

Over 110 ns simulation, distance analyses were carried about between the centre-of-mass of the ON groups of each spin label. After this, targeted MD was carried out on the SecY subunit to sample the conformational arrangement of 3DIN SecY (chain C [*Zimmer et al., 2008*]). Force constants were applied to the CA atoms, and increased at 100 ps intervals from 10 to 1000 $kJ^{-1}$ $mol^{-1}$ $nm^{-2}$

restraints over 1 ns. Following this, the systems were simulated for a further 70 ns, with the positional restraints eased to 10 kJ$^{-1}$ mol$^{-1}$ nm$^{-2}$. This therefore allowed us to simulate a SecA-bound state of the complex, without needing SecA present. Upon opening, the MTS label was able to flip into the centre of SecY. This process was modelled through a manual switching of rotamer state for the backbone cysteine. Again, distance analyses were performed between the spin labels.

## Modelling a defective signal sequence

To investigate the effect of signal sequence interaction on the system, we modelled in a known defective signal sequence (*Hizlan et al., 2012*; *Emr et al., 1980*), where a conserved four hydrophobic residue stretch is removed (here, KKTA**IAIA**VALAGFATVAS). We modelled this in PyMOL from a 1 μs snapshot of the ATP-bound SecA-SecYE-PP system. To reduce input bias, we allowed the N-terminal KK to remain in contact with the lipid phosphate groups and simply removed the four residues from the protein. We then repositioned the flanking residues slightly to allow bond formation, and relaxed the system using energy minimization. We then simulated the complex out to >400 ns for further analyses.

## Secondary structure formation analyses

Formation of secondary structure was determined both using visual analysis and quantified using the dictionary of secondary structure of proteins (DSSP) algorithm (*Kabsch and Sander, 1983*; *Joosten et al., 2011*). This was implemented using the Gromacs utility do_dssp, which provides an estimation of secondary structure for each reside based on dihedral angle, including 'no structure', 'α-helix', '3$_{10}$-helix' and 'hydrogen-bonded turn'.

## Extending simulations to investigate secondary structure formation

To provide broad conformational sampling and reduce any input bias from the starting configurations, multiple simulations were run from different starting coordinates. For this, new systems were modelled based on snapshots of the ADP- and ATP-bound systems at 1 μs. To abolish all the secondary structure within the pre-protein, the nine residues on each side of the central constriction were substituted with glycine using SCWRL4 (*Krivov et al., 2009*). Following steepest descents energy minimization and 1 ns equilibration, production runs were carried out until all of the secondary structure within this region had been abolished, according to DSSP analysis (*Joosten et al., 2011*); ~110 ns for the ATP-bound complex and ~80 ns for the ADP-bound complex (*Figure 2—figure supplement 3*). For each simulation, three time points (70, 80 and 110 ns for ATP and 50, 65 and 80 ns for ADP, with only the 50 and 65 ns simulations run for the tandem sequence (see below) simulations) were chosen which displayed no structure at all according to DSSP analysis (*Figure 2—figure supplement 3*). This represents a reasonably broad configurational sampling, and prevents the analyses being too biased by a specific starting configuration.

For each time point, an 18 residue region of pre-protein through the SecY channel was modelled in one of two different ways, with the first nine residues (ELERQHTFA) duplicated either in a mirrored (ELERQHTFAAFTHQRELE) or tandem (ELERQHTFAELERQHTFA) manner, using SCWRL4 (*Krivov et al., 2009*). These were then minimised using steepest descents, equilibrated for 1 ns and simulated for between 90 and 110 ns. The final 75 ns of each simulation run were then analysed for secondary structure. Simulations were run on EPCC's Cirrus HPC Service.

Due to sampling restrictions, we were realistically only able to analyse a single nine residue sequence. We opted to use the sequence from the original crystal structure, as this has already been shown to be stable in the SecY channel, thereby reducing the potential for artefacts arising from changes of sequence. Additionally, the sequence has already been shown to fold within a reasonable time frame (*Figure 2—figure supplement 2B*).

## Water dynamics simulations

To model the water dynamics in and around the translocon, a series of simulations were run using structural snapshots of the 1 μs SecA-SecYE-PP ATP simulation from *Figure 2* as starting points (500, 502, 504, 506, 508, 510, 600, 602, 604, 606, 608, 610, 700, 702, 704, 706, 708, 710, 800, 802, 804, 806, 808, 810, 900, 902, 904, 906, 908, 910, 1000 ns). Water dynamics simulations were run in the NVE ensemble, meaning a constant number of particles, volume and energy were maintained

throughout. The advantage of this is to avoid introduction of artificial perturbations through use of a thermostat or barostat. Simulations were run with a 1 fs time step, writing coordinates every 5 fs. The Verlet cutoff scheme was used with a buffer size of 0.001 kJ mol$^{-1}$ ps$^{-1}$ to achieve proper energy conservation, as monitored through following the energy of the simulations.

## Mean squared displacement calculations

To model the translational dynamics of the waters in the system, mean squared displacement (MSD) calculations were employed. MSD is a common statistical mechanics measure of translational motion. It follows the progression of an atom in relation to a reference position as a product of time, revealing the extent of its exploration of 3D space. Analyses were carried out as described previously (*Capponi et al., 2015*). Briefly, a box of 5×5×12 nm was built around SecY, with the geometric centre at the pore ring. This box was subdivided into 24 slices 0.5 nm thick, and the waters from each slice were analysed separately. The MSD of the waters in each slice was plotted, and the last 25 ps was fitted to power law equations (*Equation 1*).

  *Equation 1*.

$$MSD_{(t)} = kt^A \tag{1}$$

Where $k$ is a fitting parameter, and the exponent $A$ provides information on the molecular diffusivity. From this, the $A$ value was plotted for each slice.

## HOLE

To analyse the SecY cavity sizes, snapshots were taken from the 1 μs SecA-SecYE-PP ATP and ADP simulations at 500, 502, 504, 506, 508, 510, 600, 602, 604, 606, 608, 610, 700, 702, 704, 706, 708, 710, 800, 802, 804, 806, 808, 810, 900, 902, 904, 906, 908, 910 and 1000 ns. In addition, snapshots were taken at the same time points from simulations previously published (*Allen et al., 2016*) of the SecA-SecYEG complex without pre-protein, with ATP or ADP bound. Snapshots were also taken from simulations of *T. thermophilus* SecYEG (*Tanaka et al., 2015*) in a resting and open conformation (see Targeted MD section above).

For each snapshot, the centre of the SecY channel was initially defined as the geometric centre of the pore ring residues, and this was used to seed the HOLE calculations. The correct siting of the cavity was determined with visual inspection using VMD (*Humphrey et al., 1996*). For the successful calculations, data were extracted for 6.5 Å on either side of the pore ring. The area under the curves were computed and integrated using the trapezoidal rule with partitions of 0.5 Å.

## EPR DEER

Proteins for spin-labelling were produced as described previously (*Collinson et al., 2001*). Complexes were labelled by incubation with a 20 fold excess of MTS and 40 μM cardiolipin at room temperature for 30 min, in buffer A (20 mM Tris pH 8, 130 mM NaCl, 10% glycerol with 0.02% by volume n-Dodecyl-β-D-Maltoside [DDM]). Following this, excess (unbound) cardiolipin and MTS were removed using size exclusion chromatography with a superose 6 10/300 column at 4°C. Purified fractions were concentrated and stored at −80°C. SecA was purified as described previously (*Gold et al., 2007*) and stored in buffer B (20 mM Tris pH 8, 50 mM KCl, 2 mM MgCl$_2$).

The samples for EPR were prepared using these stock solutions. The proteins were mixed and diluted into the deuterated buffer C (20 mM Tris pH 8, 50 mM KCl, 2 mM MgCl$_2$ in D$_2$O) with the required amount of DDM to keep the concentration at 0.02% and MgCl$_2$ was kept constant at 2 mM. SecYEG was added to 10 μM, with SecA in 5-fold excess and nucleotide (ADP or the non-hydrolysable AMPPNP) added to 2 mM final concentration. Glycerol-$d_8$ 30% by volume was added to give a final volume of 60 μl.

The mixtures were loaded into 3 mm OD suprasil EPR tubes and flash frozen. The samples were loaded into the high-power Q-band (34 GHz, ER 5106QT-2w cylindrical resonator) Bruker Elexsys E580 spectrometer at the University of St Andrews.

The double electron electron resonance (DEER) experiments were carried out at 50 K with a four pulse, deadtime free sequence (*Milov et al., 1981*; *Martin et al., 1998*; *Jeschke, 2012*). The pump pulse (14 ns) was set to the maximum of the echo detected field swept spectrum of the nitroxide. The observer sequence was set 80 MHz offset from the pump frequency and used 32 ns pulses. The

shot repetition time was optimised to measure at least 70% of the echo height (3 ms) and for each scan, 30 shots were taken at each time point. Furthermore, a two-step phase cycle with nuclear modulation averaging (5 steps of 24 ns separation with an initial separation of 200 ns) was applied. Using DeerAnalysis2016, the data had 800 ns cut to remove artefacts at the end of the data, with a 129 ns zerotime. It was found necessary to use a fifth-order polynomial background correction (*Jeschke et al., 2006*). Distance distributions were calculated by Tikhonov Regularisation with the Regularisation parameter determined by the programme.

Note that for a fully spin-labelled SecYEG, the modulation depth would be expected to be about 0.3 rather than values in the approximate range 0.1 to 0.15 (*Figure 5E*) and thus the sample may not have been fully labelled. The differences between conditions were not experimentally reproducible with alternative samples and therefore may be due to small changes in the experimental conditions. The higher frequency of the measured dipole-dipole coupling in the case of spin-labelled SecYEG with SecA and AMPPNP indicates an alteration of the cavity dimensions. More details and experiments are shown in the Supporting Information (*Figure 5—figure supplement 2*).

The research data supporting the DEER data can be accessed at https://doi.org/10.17630/0fedaeec-7e27-4876-a6d1-cda2d3a6799c

## Hydrogen deuterium exchange mass spectrometry (HDX-MS)

Peptide identification and peptide coverage were optimized for SecYEG and SecA using un-deuterated controls. The sample workflow for HDX-MS involved mixing 10 µM of SecYEG and 15 µM of SecA and incubating for 10 min on ice. 5 µl of SecYEG-SecA complex was diluted into 95 µL of equilibration buffer C (20 mM Tris pH 8, 50 mM KCl, 2 mM MgCl$_2$ and 0.02% DDM in H$_2$O) or with deuterated buffer C (20 mM Tris pH 8, 50 mM KCl, 2 mM MgCl$_2$ and 0.02% DDM in D$_2$O) at 25°C. For experiments analysing the dynamics of the SecYEG-SecA in presence of either AMPPNP or ADP, 1 mM of nucleotide was added to the protein mixture and to buffer A.

The deuterated samples were then incubated for 0.25, 1, 5, and 30 min at 25°C. All samples were quenched with 100 µL of quench buffer (0.8% formic acid and 0.1% DDM). The mixed solution was at pH 2.4. Peptide digestion was then performed on-line using a Enzymate online digestion column (Waters) in 0.1% formic acid in water at 20°C and with a flow rate of 200 µL/min. The pepsin column was washed with pepsin wash recommended by the manufacturer (0.8% formic acid, 1.5 M Gu-HCl, 4% MeOH). To prevent significant peptide carry-over from the pepsin column, a blank run was performed between sample runs.

Peptide fragments were then trapped using an Acquity BEH C18 1.7 µM VANGUARD chilled pre-column for 3 min. Peptides were then eluted into a chilled Acquity UPLC BEH C18 1.7 µM 1.0 × 100 mm column using an 8–40% gradient of 0.1% formic acid in acetonitrile at 40 µL/min. Peptide fragments were ionized by positive electrospray into a Synapt G2-Si mass spectrometer (Waters). For high-energy acquisition of product ions, a 20 to 30 V trap collision energy ramp was used to acquire the MS$^E$ data. Mass accuracy correction was performed using Leucine Enkephalin (LeuEnk) as a lock mass and iodide was used for mass spectrometry calibration.

All deuterated time points and un-deuterated controls were carried out in triplicate. MS$^E$ data from un-deuterated controls samples of SecYEG and SecA were used for sequence identification using the Waters ProteinLynx Global Server 2.5.1 (PLGS) and filtered using DynamX (v. 3.0). Filtering parameters used were a minimum and maximum peptide sequence length of 4 and 25, respectively, minimum intensity of 1000, minimum MS/MS products of 2, minimum products per amino acid of 0.2, and a maximum MH +error threshold of 5 ppm. Furthermore, all the spectra were examined and checked visually and only peptides with a high signal to noise ratios were used for HDX-MS analysis.

## Steered MD

1 µs snapshots were taken from the SecA-SecYE-PP ADP- and ATP-bound simulations, in which four pre-protein residues in the exterior SecY cavity had formed an α-helical configuration (*Figure 2A*). From these, SecA was removed and the backbone of the pre-protein was broken at a position equivalent to proOmpA residue 25, separating the signal sequence and rest of the pre-protein. All residues after the equivalent of proOmpA residue 40 were removed (*Figure 7A*), and simulated for 5 ns. The α-helix was then either forced into an unstructured conformation using steered MD with a

pulling force of 1000 kJ mol$^{-1}$ nm$^{-1}$ for 36 ps on the C-terminal residue, or stabilised using a constraint of 0.2–0.25 nm between the backbones of the i and i + 4 helix residues, followed by five ns unbiased simulation. Then, 10 independent steered MD simulations were run for each state, where the substrate was pulled from the C-terminal residue in a z-axis direction using a force constant of 600 kJ mol$^{-1}$ nm$^{-1}$. For each repeat, the time take for the helical/non-helical region to cross the pore was measured, based on the centre-of-mass distance between the region and the SecY pore residues, Ile-78, Ile-183, Ile-275 and Ile-404 (*G. thermodentrificans* numbering).

## Acknowledgements

This work was funded by the BBSRC: BB/M003604/1, BB/I008675/1 and BB/N015126/1 to RAC, WJA and IC, and Wellcome: 104632 to IC and WJA and 109854/Z/15/Z to AP and ZA. JEL thanks the Royal Society for a University Research Fellowship and Wellcome for a Multi-User Equipment Grant (099149/Z/12/Z). TF is supported from European Regional Development Fund-Project 'Mechanisms and dynamics of macromolecular complexes: from single molecules to cells' (No. CZ.02.1.01/0.0/0.0/15_003/0000441). EP is the recipient of an Imperial College London Institute of Chemical Biology EPSRC CDT studentship. AS is funded by the EPSRC (ep/m508214/1). This work was carried out using the computational facilities of the Advanced Computing Research Centre, University of Bristol (http://www.bris.ac.uk/acrc/). Additional simulations were carried out using computer time on EPCC's Cirrus HPC Service (https://www.epcc.ed.ac.uk/cirrus) and on the ARCHER UK National Supercomputing Service (http://www.archer.ac.uk), provided by HECBioSim, the UK High End Computing Consortium for Biomolecular Simulation (http://www.hecbiosim.ac.uk/), supported by the EPSRC.

## Additional information

### Funding

| Funder | Grant reference number | Author |
| --- | --- | --- |
| Biotechnology and Biological Sciences Research Council | BB/M003604/1 | Robin A Corey<br>Ian Collinson |
| Wellcome | 109854/Z/15/Z | Zainab Ahdash<br>Argyris Politis |
| Wellcome | 099149/Z/12/Z | Anokhi Shah<br>Janet E Lovett |
| Engineering and Physical Sciences Research Council | ep/m508214/1 | Anokhi Shah |
| Biotechnology and Biological Sciences Research Council | BB/I008675/1 | William J Allen<br>Ian Collinson |
| Wellcome | 104632 | William J Allen<br>Ian Collinson |
| European Regional Development Fund | CZ.02.1.01/0.0/0.0/15_003/0000441 | Tomas Fessl |
| Royal Society | University Research Fellowship | Janet E Lovett |
| Biotechnology and Biological Sciences Research Council | BB/N015126/1 | Ian Collinson |

The funders had no role in study design, data collection and interpretation, or the decision to submit the work for publication.

### Author contributions

Robin A Corey, Conceptualization, Data curation, Formal analysis, Validation, Investigation, Visualization, Methodology, Writing—original draft, Project administration, Writing—review and editing; Zainab Ahdash, Data curation, Formal analysis, Investigation, Visualization, Writing—review and editing; Anokhi Shah, Data curation, Formal analysis, Investigation, Writing—review and editing; Euan Pyle,

Tomas Fessl, Formal analysis, Investigation; William J Allen, Conceptualization, Supervision, Writing—original draft, Writing—review and editing; Janet E Lovett, Argyris Politis, Supervision, Funding acquisition, Writing—original draft, Writing—review and editing; Ian Collinson, Conceptualization, Supervision, Funding acquisition, Writing—original draft, Project administration, Writing—review and editing

**Author ORCIDs**
Robin A Corey ![ORCID] http://orcid.org/0000-0003-1820-7993
Zainab Ahdash ![ORCID] http://orcid.org/0000-0002-4495-8689
Euan Pyle ![ORCID] http://orcid.org/0000-0002-4633-4917
William J Allen ![ORCID] https://orcid.org/0000-0002-9513-4786
Tomas Fessl ![ORCID] http://orcid.org/0000-0001-6969-4870
Janet E Lovett ![ORCID] http://orcid.org/0000-0002-3561-450X
Argyris Politis ![ORCID] http://orcid.org/0000-0002-6658-3224
Ian Collinson ![ORCID] http://orcid.org/0000-0002-3931-0503

**Decision letter and Author response**
Decision letter https://doi.org/10.7554/eLife.41803.sa1
Author response https://doi.org/10.7554/eLife.41803.sa2

## Additional files

**Supplementary files**
• Transparent reporting form

**Data availability**
All data generated during this study are included in the Figures of the mansucript. EPR data is available at https://doi.org/10.17630/0fedaeec-7e27-4876-a6d1-cda2d3a6799c.

The following dataset was generated:

| Author(s) | Year | Dataset title | Dataset URL | Database and Identifier |
|---|---|---|---|---|
| Corey RA, Ahdash Z | 2018 | EPR data from ATP-induced asymmetric pre-protein folding as a driver of protein translocation through the Sec machinery | https://doi.org/10.17630/0fedaeec-7e27-4876-a6d1-cda2d3a6799c | University of St Andrews, 10.17630/0fedaeec-7e27-4876-a6d1-cda2d3a6799c |

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
