## [Decision Letter]

[Editors’ note: a previous version of this study was rejected after peer review, but the authors submitted for reconsideration. The first decision letter after peer review is shown below.]

Thank you for submitting your work entitled "ATP-induced asymmetric pre-protein folding: a driver of protein translocation?" for consideration by *eLife*. Your article has been reviewed by two peer reviewers, and the evaluation has been overseen by a Reviewing Editor and a Senior Editor. The reviewers have opted to remain anonymous.

Our decision has been reached after consultation between the reviewers. The individual unedited reviews are provided below, and you will see that we found there to be much of value in the work, but that some serious reservations were also expressed. Based on these discussions and the individual reviews below, we regret to inform you that your work will not be considered further for publication in *eLife*.

The authors observed different channel shape of SecA bound with ATP vs. ADP from MD simulations and suggested that this should translate into different helix formation of the translocating polypeptide, which may help driven the translocation process and maintain its forward direction. While recognizing the potential importance of the model, we are concerned of the absence in the simulations of the negatively-charged lipids that are known to be crucial to the proper function of SecA. During the discussion between the reviewers and the editor, this point was identified as the most crucial weakness. Given the strengths we see in the manuscript, we encourage resubmission if this concern can be clearly addressed in a substantive manner through additional simulations.

*Reviewer #1:*

The authors previously proposed a probabilistic model for SecA-mediated protein translocation that relies on ATP-regulated gating of the SecYEG channel. This probabilistic model had several advantages but fails to explain why translocation proceeds in the forward direction. The authors provide a plausible model for how this might happen based on MD simulations of changes in the shape of the channel in the ATP- vs. the ADP-bound states of SecA. The idea (as far as I understand it) is that the 2HF decreases the size of cavity on the cytoplasmic side of the central pore ring. This biases the polypeptide to fold in the cavity on the outside of SecYEG. It is also possible that restricting the conformational entropy on the cytoplasmic face biases movement toward the exterior in order to maximise the conformational entropy of the unfolded region of the protein.

I like the model, and it sounds plausible. My concern is that the evidence is entirely based on modelling. I'm not familiar enough with MD simulations to know how accurate predictions from them might be. Thus, my prejudice is to regard this as an interesting and plausible theory without any experimental evidence. Is there any way to provide experimental evidence for this model (e.g. by artificially increasing the size of the cavity on the cytoplasmic face)?

*Reviewer #2:*

Beginning with an x-ray structure of an engineered version of SecA bound to SecYEG, the authors have carried out an extensive set of microsecond-scale simulations of SecA-SecYEG with SecA in an ATP- or ADP-bound state. They report a nucleotide-dependent asymmetry in the secondary structure of the pre-protein in the cytoplasm-facing and periplasm-facing chambers of SecYEG. They conclude that pre-protein transport is driven in part by secondary structure formation.

There are a number of problems with this paper:

1) Absence of negatively charged lipids. The simulations were carried using POPC without the presence of negatively charged lipids, such as POPG. It is well established that negative lipids in the membrane are required for SecA to function properly. There is no mention of this in the paper and no justification of using POPC alone is given.

2) Time scale problems. The literature is a bit hazy on the subject of pre-protein transport rates, which is not discussed or alluded to in the paper. The most direct reference on the subject I know of is Tomkiewicz et al. (2006). Those authors report ~270 amino acids per minute (4.5 per second). Remarkably, this is the same order of magnitude as for translocon-guided insertion (about 20 AAs/sec). In any case, despite pretty much state-of-the-art simulation times (microseconds), they are far too short compared to the physiological situation. Although the RMSDs seem small on the microsecond time scale, it is apparent that only SecYEG is really stable. The RMSDs of other portions of the protein that include SecA are steadily climbing. Overall, it is just not clear whether changes observed for SecA with bound ATP or ADP are meaningful or just hopeful on the time scale used.

3) Vague description of the building and testing of the initial structural model. In the Li et al. (2016) structure, a pseudo pre-protein was covalently linked to the so-called two-helix finger and the pre-protein was cross-linked to the SecY plug domain. While the Li et al. structure is an important contribution, one must be cautious about structures that are made to look like one's hopeful cartoons. In the Li et al. structure, many residues were not resolved in both SecY and SecA. Consequently, these were modeled into the simulation structure. It is not clear how the model was equilibrated or tested before serious simulations were run. Overall, I have little confidence in the starting model based upon the description the authors provide.

4) Comparative examination of SecA and SecY. There are lots of SecA structures. The first thing I would want to do is to compare the structure of the model SecA in the SecA/SecY complex with known structures. Ditto for SecY. Of particular importance for the present paper is a comparison of the ATP- and ADP-bound states of SecA. While the authors look closely at what goes in the heart of the translocon, they say very little about structural changes in SecA that are supposed underlie the changes seen in SecY.

[Editors’ note: what now follows is the decision letter after the authors submitted for further consideration.]

Thank you for submitting your article "ATP-induced asymmetric pre-protein folding as a driver of protein translocation through the Sec machinery" for consideration by *eLife*. Your article has been reviewed by two peer reviewers, and the evaluation has been overseen by a Reviewing Editor and John Kuriyan as the Senior Editor. The reviewers have opted to remain anonymous.

The reviewers have discussed the reviews with one another and the Reviewing Editor has drafted this decision to help you prepare a revised submission.

Summary:

Using molecular dynamics simulations, the authors convincingly show that the SecYEG-SecA complex undergoes significant conformational changes upon ATP binding and hydrolysis, which expands the exterior cavity of SecYEG and induces secondary structure formation of the nascent protein therein. The formation of secondary structure is linked to lowered rates of backtracking in the translocon. This suggests a molecular mechanism for how ATP hydrolysis in SecA can drive directional transport of a nascent protein through. This work may represent an important conceptual advance in our understanding of the mechanism of protein translocation by SecA-SecYEG.

Essential revisions:

The reviewers suggested acceptance of this manuscript with the following revisions:

1) The authors plausibly suggest that it is more difficult for folded structures – even α-helices – than unfolded structures to pass through SecYEG. It is not made sufficiently clear, however, whether the conformational changes in the SecA-SecYEG complex from ATP hydrolysis promote folding of the pre-protein on the exterior face of SecYEG, or unfolding of the pre-protein in the interior cavity of SecYEG, or both.

2) The section describing the solvent (dis)order (Figure 3) is somewhat confusing. Is the chemical or physical basis for the increase in disorder known? Because the authors specifically mention this effect, and because the effect is so pronounced, it should be discussed in the Discussion as to how it affects folding of the pre-protein.

3) The final (diagrammatic) main figure seems unnecessarily complicated and overly reliant on the legend. (Does it really need 11 cartoon states connected with arrows?) Simplifying this figure is important to better conveying the key conclusions to the readers.

4) Although this is not a precondition for the acceptance, the reviewers think the manuscript may benefit from additional simulations in which the nascent chain is pulled from the exterior side of the channel with SecYEG in the two different conformations induced by SecA (ATP-bound and ADP-bound).

---

## [Author Response]

[Editors’ note: the author responses to the first round of peer review follow.]

The authors observed different channel shape of SecA bound with ATP vs. ADP from MD simulations and suggested that this should translate into different helix formation of the translocating polypeptide, which may help driven the translocation process and maintain its forward direction. While recognizing the potential importance of the model, we are concerned of the absence in the simulations of the negatively-charged lipids that are known to be crucial to the proper function of SecA. During the discussion between the reviewers and the editor, this point was identified as the most crucial weakness. Given the strengths we see in the manuscript, we encourage resubmission if this concern can be clearly addressed in a substantive manner through additional simulations.

Thank you, this was a great suggestion and ties in very nicely with our recently published SecYEG-lipid interaction paper (Corey et al.,2018). So, we have been able to add an extensive amount of simulation data to address this point. Firstly, we have carried out μs simulations using a coarse-grained molecular force field of the protein complexes in a mixed bilayer, with PE, PG and cardiolipin present (Figure 2—figure supplement 4C-D). This permitted not only analysis of the effects of acidic lipids in the membrane, but also the effect of specific Sec-acidic lipid interactions (as we discussed in Corey et al., 2018).

We then carried out a new tranche of folding simulation data using these systems, to assay the effect of acidic lipids on the observed pre-protein folding asymmetry. In total this amounts to 2 μs of additional atomistic simulation – achievable thanks to a generous provision of resources from EPCC’s cirrus.

The new folding data (Figure 2—figure supplement 4E) reveals that the previously observed pre-protein asymmetry in secondary structure is not dependent of specific or general interactions with acidic lipids.

Reviewer #1:The authors previously proposed a probabilistic model for SecA-mediated protein translocation that relies on ATP-regulated gating of the SecYEG channel. This probabilistic model had several advantages but fails to explain why translocation proceeds in the forward direction. The authors provide a plausible model for how this might happen based on MD simulations of changes in the shape of the channel in the ATP- vs. the ADP-bound states of SecA. The idea (as far as I understand it) is that the 2HF decreases the size of cavity on the cytoplasmic side of the central pore ring. This biases the polypeptide to fold in the cavity on the outside of SecYEG. It is also possible that restricting the conformational entropy on the cytoplasmic face biases movement toward the exterior in order to maximise the conformational entropy of the unfolded region of the protein.

The reviewer has given a fair description of the model, although we see that the cytoplasmic cavity actually increases, rather than decreases. We have amended the relevant sections of text to read more clearly.

I like the model, and it sounds plausible. My concern is that the evidence is entirely based on modelling. I'm not familiar enough with MD simulations to know how accurate predictions from them might be. Thus, my prejudice is to regard this as an interesting and plausible theory without any experimental evidence. Is there any way to provide experimental evidence for this model (e.g. by artificially increasing the size of the cavity on the cytoplasmic face)?

The reviewer is correct, and it is indeed more satisfying to present laboratory based empirical support.

We have now significantly bolstered our MD simulations, obtaining a large amount of additional data in the presence of acidic lipids, which was the primary condition for resubmission. To ensure that this was done as fully as possible, we built on analyses we recently published in PNAS (Corey et al., 2018), using coarse-grained simulation to identify multiple Sec-lipid interactions (Figure 2—figure supplement 4C-D). We then analysed the systems as per the rest of the data, incorporating it into the main analysis (Figure 2C) as well as providing a comparative analysis (Figure 2—figure supplement 4E).

In addition, we have invested considerable effort, hence the long delay, to satisfy the additional concerns of the reviewers; even though their resolution was not a strict condition for the resubmission. We have added two sets of empirical observations on the *E. coli* SecA-SecYEG complex. The new experiments probe the nature of the asymmetrical cavities in the ATP state with combined EPR DEER spectroscopy and additional MD simulations (Figure 5 and supplemental panels), as well as state-of-the art HDX/MS (Figure 6 and supplemental panels). We believe the results add valuable support to the suppositions based on our calculations.

Reviewer #2:Beginning with an x-ray structure of an engineered version of SecA bound to SecYEG, the authors have carried out an extensive set of microsecond-scale simulations of SecA-SecYEG with SecA in an ATP- or ADP-bound state. They report a nucleotide-dependent asymmetry in the secondary structure of the pre-protein in the cytoplasm-facing and periplasm-facing chambers of SecYEG. They conclude that pre-protein transport is driven in part by secondary structure formation.There are a number of problems with this paper:1) Absence of negatively charged lipids. The simulations were carried using POPC without the presence of negatively charged lipids, such as POPG. It is well established that negative lipids in the membrane are required for SecA to function properly. There is no mention of this in the paper and no justification of using POPC alone is given.

The reviewer is correct in that acidic lipids, particularly cardiolipin (CDL), are important to Sec function. We chose to use a standard POPC membrane model as it was not clear exactly what the role of these lipids may be. However, since then we have reported on the nature of Sec-lipid interaction (Corey et al.,2018) so can now address this issue.

We have addressed this point and included the negative phospholipids (PG and CDL) in the simulations (see response to reviewer 1).

2) Time scale problems. The literature is a bit hazy on the subject of pre-protein transport rates, which is not discussed or alluded to in the paper. The most direct reference on the subject I know of is Tomkiewicz et al. (2006). Those authors report ~270 amino acids per minute (4.5 per second). Remarkably, this is the same order of magnitude as for translocon-guided insertion (about 20 AAs/sec). In any case, despite pretty much state-of-the-art simulation times (microseconds), they are far too short compared to the physiological situation. Although the RMSDs seem small on the microsecond time scale, it is apparent that only SecYEG is really stable. The RMSDs of other portions of the protein that include SecA are steadily climbing. Overall, it is just not clear whether changes observed for SecA with bound ATP or ADP are meaningful or just hopeful on the time scale used.

On the whole, SecA is stable during these simulations. We highlight the less-stable region, which has to be extensively remodeled from the crystal structure. This is an unfortunate consequence of the modelling process, but – owing to the high quality of data for SecY and the SecA 2HF, and the distance of this region from the center of the channel – the chance of it producing artefacts is very low. This is especially the case as one of the key observations, the asymmetric cavity sizes, is visible in multiple simulations from different starting coordinates, as well with additional experimental data.

With respect to the timescale, the reviewer is correct in that we are sampling far shorter time scales than physiological translocation rates. That said, we are able to sample a very kinetically fast event – the asymmetric formation of pre-protein structure – which could undoubtedly have a profound effect on the pre-protein in the ms timescale. We have added a short passage in the Discussion to address this concern.

3) Vague description of the building and testing of the initial structural model. In the Li et al. (2016) structure, a pseudo pre-protein was covalently linked to the so-called two-helix finger and the pre-protein was cross-linked to the SecY plug domain. While the Li et al. structure is an important contribution, one must be cautious about structures that are made to look like one's hopeful cartoons. In the Li et al. structure, many residues were not resolved in both SecY and SecA. Consequently, these were modeled into the simulation structure. It is not clear how the model was equilibrated or tested before serious simulations were run. Overall, I have little confidence in the starting model based upon the description the authors provide.

The reviewer is correct to be concerned, as the original crystal structure was limited in a number of ways. We have considerably extended our description of the modelling process to address these concerns.

4) Comparative examination of SecA and SecY. There are lots of SecA structures. The first thing I would want to do is to compare the structure of the model SecA in the SecA/SecY complex with known structures. Ditto for SecY. Of particular importance for the present paper is a comparison of the ATP- and ADP-bound states of SecA. While the authors look closely at what goes in the heart of the translocon, they say very little about structural changes in SecA that are supposed underlie the changes seen in SecY.

This is a good idea, however, as in our previous study (Allen et al., 2016), there are only very minor changes in SecA during the hydrolysis cycle. The only substantial changes appear to be in the SecA loop, which are most likely a result of the modelling process and likely do not affect the data here. Therefore, we do not include a comparison of the SecA models.

However, we have added initial MD data providing structural conformations of two different structures of SecYEG: 5AWW and 3DIN. We analyse these in the context of the other data, and in the general findings of the paper.

In addition, we have made the following major figure changes:

Main figures:

– Figure 1 remade to be visually clearer, and introduce icon style

– Figure 2 clarified, with additional analyses added in panel B, additional data added to panel C, and panels C-E made clearer with respect to each other. Some data brought in from supplement panels

– Figure 3 clarified with additional labels, panel C removed

– Minor adjustments to Figure 4

– Figure 5 (and supplements) – whole new data set, including new MD simulations (panels A-D) and EPR DEER data (panels E-F)

– Figure 6 – whole new data set, incorporating extensive HDX/MS data

– Figure 8 – model remade with new icon style and incorporating additional stages

Supplementary figures:

– Figure 2—figure supplement 1 – replotted on same y axes for clarity

– Figure 2—figure supplement 3 – additional analyses in panel A

– Figure 2—figure supplement 4 – panel A removed as confusing, additional panels to address role of acidic lipids in system (panels C-E)

– Figure 3—figure supplement 3 – removed as unclear

– Figure 3—figure supplement 4 – removed as unclear

– Figure 4—figure supplement 1 – removed as not necessary

– Figure 4—figure supplement 2 – panels B and C removed as superseded by new experimental analyses

– Figure 5—figure supplements 1-2 – additional figures in support of Figure 5

– Figure 6—figure supplement 1 – additional figure in support of Figure 6

[Editors' note: the author responses to the re-review follow.]

Essential revisions:The reviewers suggested acceptance of this manuscript with the following revisions:1) The authors plausibly suggest that it is more difficult for folded structures – even α-helices – than unfolded structures to pass through SecYEG. It is not made sufficiently clear, however, whether the conformational changes in the SecA-SecYEG complex from ATP hydrolysis promote folding of the pre-protein on the exterior face of SecYEG, or unfolding of the pre-protein in the interior cavity of SecYEG, or both.

From the data, we believe that it is prevention of folding on the cytoplasmic side of the channel which is driving transport. This distinction is not clear from the pre-protein folding data, and only really comes out from the cavity size analyses.

This makes the point slightly uncertain and, as it doesn’t really change the nature of the model, we choose not to state it too strongly either way. However, the reviewers are right to say that we need clarification of the data, so we have added a section to the Results to better highlight our interpretation of the data.

2) The section describing the solvent (dis)order (Figure 3) is somewhat confusing. Is the chemical or physical basis for the increase in disorder known? Because the authors specifically mention this effect, and because the effect is so pronounced, it should be discussed in the Discussion as to how it affects folding of the prerotein.

We appreciate the reviewers’ confusion, as we are unable to draw absolute conclusions from the data. We have added a paragraph to the Results section clarifying our understanding of the data.

3) The final (diagrammatic) main figure seems unnecessarily complicated and overly reliant on the legend. (Does it really need 11 cartoon states connected with arrows?) Simplifying this figure is important to better conveying the key conclusions to the readers.

We agree with the reviewers and have substantially simplified and improved this figure.

4) Although this is not a precondition for the acceptance, the reviewers think the manuscript may benefit from additional simulations in which the nascent chain is pulled from the exterior side of the channel with SecYEG in the two different conformations induced by SecA (ATP-bound and ADP-bound).

We have added an additional data set to this extent (incorporated into Figure 7).